



# Changes of the tropical glaciers throughout Peru between 2000 and 2016 – Mass balance and area fluctuations

Seehaus Thorsten[1], Malz Philipp[1], Sommer Christian[1], Stefan Lippl[1], Alejo Cochachin[2],Matthias Braun[1]

[1]Institute of Geography, Friedrich-Alexander-University Erlangen-Nuremberg, Wetterkreuz 15, 91058 Erlangen, Germany

[2]Unidad de Glaciología y Recursos Hídricos (UGRH), Autoridad Nacional del Agua (ANA), 02001 Huaraz, Perú

*Correspondence to*: Thorsten Seehaus (thorsten.seehaus@fau.de)

**Abstract.** Glaciers in tropical regions are very sensitive to climatic variations and thus strongly affected by climate change. The majority of the tropical glaciers worldwide are located in the Peruvian Andes, which have shown significant ice loss in the last century. Here, we present the first multi-temporal, region wide survey of geodetic mass balances and glacier area

fluctuations throughout Peru covering the period 2000-2016. Glacier extents are derived from Landsat imagery by performing automatic glacier delineation based on a combination of the NDSI and band ratio method and final manual inspection and correction. A total glacier area loss of -548.5±65.7 km² (-29%, -34.3 km² a⁻¹) is obtained for the study period. Using interferometric satellite SAR acquisitions, bi-temporal geodetic mass balances are derived. An average specific mass balance of -357±43 kg m⁻² a⁻¹ is found throughout Peru for the period 2000-2016. However, there are strong regional and

temporal differences in the mass budgets ranging from 68±102 kg m⁻² a⁻¹ to -990±476 kg m⁻² a⁻¹. The ice loss increased towards the end of the observation period. Between 2013 and 2016, a retreat of the glaciated area of -203.8±65.7 km² (-16%, -101.9 km² a⁻¹) is mapped and the average mass budget amounts to -836±188 kg m⁻² a⁻¹. The glacier changes revealed can be attributed to changes in the climatic settings in the study region, derived from ERA-Interim reanalysis data and the Oceanic Niño Index. The intense El Niño activities in 2015/16 are most likely the trigger for the increased change rates in the time

interval 2013-2016. Our observations provide fundamental information on the current dramatic glacier changes for local authorities and for the calibration and validation of glacier change projections.

## 1     Introduction

Tropical glaciers in the Peruvian Andes are very sensitive to climate change and rapidly respond to varying climate settings (e.g. Kaser and Osmaston, 2002; Rabatel et al., 2013). A marked decrease in glacier coverage in Peru has be reported by various studies (e.g. Georges, 2004; Hanshaw and Bookhagen, 2014; Vuille et al., 2008) for the last decades. The recession

of the Peruvian glaciers is proposed to have significant impact on the downstream ecosystem and communities (Vuille et al., 2018). Glaciers act as an important temporal water reservoir for precipitation during the wet season. Glacier meltwater runoff buffers the water shortage caused by the low precipitation during the dry season (Kaser et al., 2003; Schauwecker et al., 2017). The shrinkage of glaciers leads to higher meltwater discharge and thus increases water supply to streamflow. However, glacier runoff decreases after the glacier loss reaches a critical transition point (Pouyaud et al., 2005). It has been

suggested that some watersheds in the Cordillera Blanca have crossed already this critical transition point (Baraer et al., 2012). An unsteady or unreliable water runoff is known to cause several socio-economic issues. Hydropower production and mining rely on a continuous water supply. Moreover, glacier runoff is an important water resource for irrigation and has the potential to affect large-scale but also subsistence agriculture (Vuille et al., 2018). It also impacts the Andean ecosystems.





For example, the *bofedales* (high-altitude wet lands in the Andes) are very sensitive to changes in glacier runoff and a
depletion of the meltwater is likely to cause them to shrink (Polk et al., 2017). Additionally, the glacier retreat leads to the
formation and extension of pro-glacial lakes (Hanshaw and Bookhagen, 2014; Lopez et al., 2010) threatening downstream
areas due to their potential to cause glacier lake outburst floods (GLOF). In the Cordillera Blanca, several GLOFs have
harmed local communities in the past. The most dramatic was the disaster in 1941 when large parts of the city of Huaraz
were destroyed by a GLOF event, leading to ~1800 casualties (Carey, 2010). However, GLOF imminences are present
throughout the Tropical Andes (Cook et al., 2016; Hoffmann, 2012).

Several studies have been carried out to map and quantify changes in the glacier area in Peru. The majority of the analyses
have focused on Peru's largest glaciated region, the Cordillera Blanca (e.g. Baraer et al., 2012; Georges, 2004; Hastenrath
and Ames, 1995; Racoviteanu et al., 2008; Silverio and Jaquet, 2005, 2017; Unidad de Glaciologia y Recursos Hidricos
(UGRH), 2010), and revealed significant glacier retreat in the last decades. The most recent studies reported a glacier
recession in the Cordillera Blanca of -46% between 1930 and 2016 (Silverio and Jaquet, 2017) and -33.5% between 1975
and 2016 (Veettil, 2018). In other glaciated regions in Peru, distinct glacier retreat was observed as well. In the second
largest glaciated mountain range in Peru, the Cordillera Vilcanota, an area loss of -32% in the period 1985-2006 (Salzmann
et al., 2013) and -30% in the period 1988-2010 (Hanshaw and Bookhagen, 2014) was revealed. The only countrywide
estimation of glacier area changes was carried out by the UGRH (2014). They estimated a total glacier retreat of -42.64%
from the first Peruvian glacier inventory of 1970 (Hidrandina SA, 1989) and the recent inventory covering the period 2003-
2010 (UGRH, 2014). So far, no multi-temporal nor recent quantification of glacier area changes throughout Peru is available,
only studies at regional levels.

There are a few studies dealing with surface elevation and ice volume/mass changes in Peru. Changes in the ice volume of -
57*10⁻⁶ m³ have been derived from aerial photographs and GPS point measurement data for three glaciers in the Cordillera
Blanca by Mark and Seltzer (2005) for the period 1962-1999. Huh et al. (2017) calculated the surface elevation changes of
six glaciers in the Cordillera Blanca by means of photogrammetric digital elevation models (DEMs) and LiDAR
measurements. They found glacier wide average surface lowering ranging between -9.5 and -64.06 m in the period 1962-
2008. In the Cordillera Vilcanota, Salzmann et al. (2013) estimated volume changes of -40 to -45% based on inventory
parameters for the period 1962-2006. The only large-scale mass balance estimates covering Peru are the following: a mass
balance estimation for the low latitudes of -1080±360 kg m⁻² a⁻¹ based on the upscaling of glaciological mass balance
measurements covering the period 2003-2009 (Gardner et al., 2013); a mass balance calculation throughout South America
(excluding Patagonia) of -6±12 Gt a⁻¹ using space borne gravimetric measurements from the Gravity Recovery and Climate
Experiment (GRACE) for the period 2000-2010 (Jacob et al., 2012); and a geodetic mass budget of -0.49±0.09 Gt a⁻¹ (-
227±42 kg m⁻² a⁻¹, ice density scenario: 850 kg m⁻³) derived from InSAR measurements for the period 2000-2012/13
including glaciers in Bolivia (Braun et al., 2019). The first two cover large areas and thus the mass balance signals of
glaciers in Peru or even smaller regions cannot be derived. The latter uses the glacier boundaries defined by the Randolph
Glacier Inventory (RGI) 6.0. The RGI 6.0 has certain limitations in this region (Section 3 and RGI Consortium, 2017), which
can lead to biases in the mass balance computation (Section 6.2).

Up to now, a spatially detailed and multi-temporal quantification of glacier changes throughout Peru is missing. In order to
address this issue, this work aims to continue and expand the glacier monitoring of previous studies by carrying out a
comprehensive analysis of glacier area changes and mass balances throughout the Peruvian Cordilleras for the observation
period 2000-2016 based on multi-sensor remote sensing data. The main objectives of this study are:

- to obtain a temporally and methodically consistent evaluation of countrywide glacier area changes





- • to assess geodetic glacier mass balances and their temporal variations throughout Peru

• to identify relations between glacier fluctuations, changes of climatic variables and topographic parameters

## 2    Study site

Peru is home to the majority of tropical glaciers worldwide. About 70% of all tropical glaciers, covering an area of 1602.96 km² (RGI 6.0), are located there. The Peruvian Andes are subdivided into three major mountain ranges, the Cordillera Occidental, Central and Oriental, from west to east, and several smaller Cordilleras (Figure 1). According to Sagredo and Lowell (2012), the glaciated areas are divided into three subregions based on their climatic settings:

• R1: Northern wet outer tropics, with a high mean annual humidity of 71%, nearly no seasonality of the temperature (annual mean: 1.6 °C) and a total annual precipitation of 815 mm. R1 ranges from the Cordillera Blanca southwards to the Cordillera Chonta and also includes the Cordilleras Huagoruncho and Huaytapallana further east.

   • R2: Southern wet outer tropics, with moderate mean annual humidity of 59%, an annual seasonality of the mean monthly temperature of about 4 °C (annual mean: 1.6 °C) and a total annual precipitation of 723 mm. R2 ranges

from the Cordillera Vilabamba westwards to the Cordillera Apolobamba (partly located in Bolivia, but included completely in this study).

   • R3: Dry outer tropics, with low mean annual humidity of 50%, a mean annual temperature of -4.0 °C (seasonality of the mean monthly temperature of ~5°C) and low total annual precipitation of 287 mm. R3 ranges from the Cordillera Ampato westward to the Cordillera Volcanica.

The annual variability of precipitation shows a strong seasonality in all three subregions (Sagredo and Lowell, 2012) with a dry season during austral winter from May to September and a wet season during austral summer from October to April. The glaciers accumulate mass almost exclusively during the wet season, whereas the lower reaches of the glaciers experience ablation throughout the year (Kaser, 2001). Thus, slight variations in precipitation and temperature can lead to strong changes of the glacier mass balances (Francou Bernard et al., 2003), but also surface albedo and radiation significantly affect

the mass budget of tropical glaciers (Favier et al., 2004; Wagnon et al., 1999). Moreover, the reaction of the glaciers in the Tropical Andes to changing environmental conditions is nearly immediate (Vuille et al., 2008). The El Niño Southern Oscillation (ENSO) has a strong impact on climate and thus the glacier mass balances in Peru. El Niño events typically lead to pronounced glacier mass losses due to an induced precipitation deficit and above average temperatures, whereas during La Niña periods the opposite conditions lead to reduced mass losses or even mass gain (Favier et al., 2004; Vuille et al., 2008).

Glaciological mass balance measurements are carried out at several glaciers in the study region by the UGRH, a subdivision of the Autoridad Nacional del Aqua (National Water Administration). However, observations are available in the World Glacier Monitoring Service (WGMS) database only for two glaciers covering our study period 2000-2016. Continues annual



mass balance observations have been documented at Artesonraju Glacier and Yanamarey Glacier since 2004. At some additional glaciers, mass balance programmes were initiated later and the data is not yet archived in the WGMS database.

## 3        Data

Spaceborne remote sensing data from different sensor systems is collected to perform this comprehensive study on glacier changes in the period 2000-2016. Synthetic Aperture Radar (SAR) data is applied to obtain information on glacier surface elevation changes and mass balances. Digital elevation models (DEM) derived from interferometric SAR acquisitions at different time steps are applied to compute surface elevation change information. As the elevation reference at the start of our observation interval, the void-fill LP DAAC NASA Version 3 SRTM DEM (NASA JPL, 2013) is used. It is based on

bistatic C-band SAR data, acquired during the Shuttle Radar Topography Mission (SRTM) by the National Aeronautics and Space Administration (NASA) and the German Aerospace Center (DLR) in February 2000. DEMs of later dates are generated from bistatic X-band SAR imagery of DLR's TanDEM-X (TDX) mission, which started in 2010 (Zink et al., 2011) (Section 6). Both SAR missions acquired data using different radar bands. Signal penetration of different radar frequencies in glacier surfaces depends on water content and the density of upper layers. This can lead to biases when comparing elevations

on glaciated areas. Thus, we tried to select only imagery from the same season as the SRTM data in order to obtain acquisitions with similar glacier surface conditions (Section 8) and to avoid seasonal mass balance biases. In early 2013, an almost complete coverage of the glaciated regions in Peru could be obtained (early 2012 at subregion R1 as well as for comparison with Braun et al., 2019). Only a small fraction of the glaciated areas in subregion R3 had no coverage by TDX in early 2013. Therefore, TDX imagery from early 2014 is used to fill the gaps (Section 9). A second temporally consistent

coverage of the Peruvian glaciers by TDX data is available for 2016, though acquired primarily in the months of October and November, which mark the end of the dry season and beginning of the wet season. Less ablation occurs during the dry season (Favier et al., 2004; Kaser, 2001; Veettil et al., 2017b). Thus, the mass balances for observation periods ending in 2016 would be more negative when considering this seasonal bias. It is difficult to adequately quantify this temporal bias. Therefore, no correction is employed in the analysis and our computed mass loss rates represent lower bound estimations for

the periods 2000-2016 and 2013-2016. A summary of the 331 analysed TDX scenes is provided in Table S1 and the spatial coverage of the subregions is plotted in Figure S1.

The RGI 6.0 Region 16 "Low Latitudes" covers all glaciated regions in Peru. The outline dates range between 2000 and 2009 in Peru. Thus, it does not represent the glacier extent at a specific moment. Moreover, it is mentioned in the RGI 6.0 Technical Report (RGI Consortium, 2017) that significant snow contamination caused difficulties in the glacier delineations,

especially in southern Peru, and that a more rigorous demarcation could decrease the total glacier area. The Peruvian glacier inventory compiled by UGRH (2014) is also not temporally consistent. It covers the period 2003-2010. In order to use temporally appropriate glacier outlines for the mass balance evaluations and to map coincidental glacier area changes, we decided to generate a consistent database of countrywide glacier extents that correlates with the dates of our coverages of



interferometric SAR data (see above). Therefore, cloud-free multispectral images from Landsat 5 TM and Landsat 8 OLI are
135 ordered from the United States Geological Survey (USGS). Imagery, preferably during the dry season, is selected to reduce
distortions due to temporal snow cover. For all subregions, a complete coverage in 2000 and 2016 is available. In subregions
R1 and R2, cloud and snow cover forced us to map a small fraction of the glaciated regions using imagery from 2014
(Section 9). An overview of the analysed Landsat imagery is presented in Table S2. The Cordillera Blanca is the only
mountain range with a considerable debris-covered glacier fraction. Therefore, the mapping of the glacier extents in this area
is supported by interferometric analysis of repeat pass SAR acquisitions from the TerraSAR-X and Sentinel-1 satellite
missions.

ERA-Interim reanalysis data (Dee et al., 2011) covering the period 1979-2017 provided by the Center for Medium-Range
Weather Forecasts (ECMWF) is used to evaluate climatic changes and to identify correlations between glacier fluctuations,
skin temperature, total precipitation and downward surface thermal radiation. Monthly Oceanic Niño Index (ONI) data is
145 applied as a proxy for ENSO events, which is available from the National Oceanic and Atmospheric Administration (NOAA)
Climate Prediction Center (http://origin.cpc.ncep.noaa.gov/products/analysis_monitoring/ensostuff/ONI_v5.php). According
to NOAA's definition, ONI values above +0.5 indicate El Niño events, whereas La Niña is present when ONI values are
below -0.5.

## 4        Methods

### 4.1        Glacier inventory

Since the manual delineation of glacier extents is laborious, time-consuming and subjective, several methods have been
developed to automatically map glacier outlines based on multispectral images (Veettil and Kamp, 2017). The most widely
used and robust approaches are the computation of the normalized difference snow index (NDSI) or the band ratio (BR) and
the application of a threshold value to differentiate between on- and off-glacier areas (GLIMS algorithm working group, n.d.;
Paul et al., 2013). In this study, we first used the NDSI to classify glacier areas and combined it with BR information to
improve the mapping in areas affected by shadows. NDSI maps generated from top of atmosphere reflectance values show a
155 better performance than NDSI maps based on digital number values. For the BR computation, digital number values are
taken. The threshold value selection is supported by high-resolution satellite imagery (Google Earth) from the respective
dates. A NDSI threshold value of 0.8 is selected, which is higher than the thresholds of 0.5-0.6 applied by other studies in
this region (e.g. Silverio and Jaquet, 2005; Veettil et al., 2017). This offset might be induced by the application of top of
atmosphere reflectance values instead of digital number values. The threshold for the BR data is set to 1.7 for Landsat 5 TM
and 1.5 for Landsat 8 OLI data. Finally, polygons of the glacier outlines are generated from the computed glacier masks.

The detection of the debris-covered glacier termini extents in the Cordillera Blanca is difficult using multi-spectral imagery.
Therefore, we generated SAR coherence maps from repeat-pass SAR acquisitions to distinguish the debris-covered ice from
the surrounding ice-free areas (Atwood et al., 2010; Lippl et al., 2018). The surface structure of the debris-covered glacier



areas changes over time due to the dynamics and melting of the underlying ice. This leads to a temporal decorrelation of the
backscattered SAR signal of repeat-pass SAR imagery and thus to lower coherence as compared to the surrounding ice-free
areas. This difference in coherence facilitates the delineation of the debris-covered ice areas. Data from the Sentinel-1
mission are used to map the debris-covered areas in 2016. No suitable repeat-pass SAR acquisitions are available for the
Cordillera Blanca in 2013. Thus, we had to rely on TerraSAR-X and Sentinel-1 data from 2014 to map the outlines of the
debris-covered glacier tongues. In 2000 (and ±1 year), only repeat-pass SAR data is available from the European Remote
Sensing (ERS) satellite with a repetition cycle of one day. Due to this short temporal baseline, a separation between debris-
covered ice and surrounding ice-free areas is unfeasible. Consequently, we combined our outlines from 2000 with the
manually delineated debris-cover masks available from the Global Land Ice Measurements from Space (GLIMS) database
from 2003 based on SPOT imagery (mapped by Adina Racoviteanu).

The catchment discriminations of the RGI are applied to split the resulting polygons into individual glacier basins. In the
next step, the glacier inventories are visually inspected and misclassified areas are manually corrected. These manual
corrections are supported by high-resolution imagery from the respective years (Google Earth). According to the RGI 6.0
Technical Report (RGI Consortium, 2017), topographic parameters (minimum, maximum and median elevation, mean slope
and aspect) of the individual glacier basins are computed using the void-filled SRTM DEM as an elevation reference.
Finally, the areas (S) of the complete inventories and of each glacier are measured in UTM projection (UTM Zone 18S for
subregion R1 and UTM Zone 19S for subregion R2 and R3). The uncertainties of the area gauging ($\delta_S$) are calculated
following the approach of Malz et al. (2018) based on an error evaluation of 3% for alpine glacier outlines derived from
Landsat images (Paul et al., 2013). This estimate is scaled by the area to perimeter ratio of the studied subregion compared to
the area to perimeter ratio of Paul et al. (2013) in order to account for differences in the shape of the glaciated areas.

### 4.2   Elevation change

Surface elevation change information is computed by differencing DEMs from SRTM and TDX data. Therefore, DEMs are
derived from the bistatic TDX imagery following the differential interferometric approach (e.g. Malz et al., 2018; Seehaus et
al., 2015; Vijay and Braun, 2016), which is briefly summarized in the following.

First, acquisitions from the same relative orbit and date are concatenated in the along track direction. A differential
interferogram is computed using the void-filled SRTM DEM as elevation reference. In the next steps, the interferogram is
filtered, unwrapped by applying the branch cut and minimum cost flow algorithm and the unwrapped differential phase is
transferred into differential elevations. Subsequently, the topographic information of the SRTM DEM is added to obtain
absolute height information and finally the product is geocoded and orthorectified. The DEMs are visually checked for
phase-jumps and the best results of both phase-unwrapping methods are selected for further processing. Areas affected by
remaining phase-jumps are masked out.

The TDX DEMs need to be precisely horizontally and vertically coregistered to the respective reference DEM (SRTM for
2000, TDX for 2013) in order to accurately map elevation changes on the glaciated areas. Figure S2 illustrates the applied





processing chain used to perform this coregistration. First, smooth stable reference areas are defined by masking out vegetation, water and glacier areas. The vegetation and water masks are derived from region wide cloud free Landsat 8 mosaics and using a normalized difference vegetation index (NDVI) threshold of 0.3 and a normalized difference water index (NDWI) threshold of 0.1. Additionally, a slope threshold of 15° (of the respective reference DEM) is applied.

Thereafter, the TDX DEMs are bi-linearly vertically corrected for offsets to the reference DEM, which are measured on the defined stable regions. Subsequently, a horizontal coregistration between the reference DEM and the TDX DEMs is carried out following the widely used approach of Nuth and Kääb (2011). Afterwards, a second bi-linear vertical coregistration of the TDX DEMs to the reference DEM is run to reduce any biases that remain. Finally, the coregistered TDX DEMs are merged to a regional DEM mosaic, which include a date stamp for each grid cell.

To obtain elevation change rates $\Delta h/\Delta t$ of the respective study periods, the SRTM DEM and the TDX DEM mosaics are differentiated. Therefore, the mean date of the eleven-day SRTM mission (2000-02-16) is assigned to the SRTM DEM. Since data voids in the SRTM DEM are filled with data from other sources (no date information available), the non-SRTM data values are masked out using the coverage information provided by LP DAAC NASA. Numerous studies have revealed that the glaciers in Peru are in general retreating (Section 1). Thus, the glacier inventory from the beginning of the respective

observation period is employed to create surface elevation change maps for on- and off-glacier areas. The average regional and glacier-wise elevation change rates are obtained by integration of $\Delta h/\Delta t$ over the respective areas. Slopes steeper than 50° are rejected (5.7% of the glacier area in 2000), since major ice aggregation is quite unlikely there (avalanche slopes, backed up by field observations) and DEMs are less accurate on these steep slopes (Toutin, 2002). To account for data voids in the elevation change fields on glaciated areas, the measured $\Delta h/\Delta t$ values are area weighted based on the hypsometric area

distribution using 100 m elevation bins. Outliers in the respective elevation bins are sorted out using three times the normalized median absolute deviation (NMAD) (Brun et al., 2017). For all hypsometric analyses of elevation changes (SRTM to TDX, TDX to TDX), the void-filled SRTM DEM is utilized.

The uncertainties of the generated elevation change rates are assessed by evaluating the elevation change rates on non-vegetated stable off-glacier areas (water and vegetation masks, see above). The lowest and highest 2% quantiles of the

change rates are rejected to suppress the impact of processing artefacts and outliers. To account for the dependency of the offsets on the slope (Figure 2 and S3 and S4), the deviations are binned in slope intervals of 5°. Remaining outliers are removed by employing a 3*NMAD filter for each slope bin. Finally, the area-weighted standard deviations $\sigma_{AW}$ based on the offsets in off-glacier areas and the slope distribution in glacier areas are calculated.

Since we integrate elevation change information over the glaciated area, spatial auto correlation of the elevation change

fields must be considered in the accuracy assessment. We estimated the uncertainty of the computed average elevation change rates ($\sigma_{\Delta h/\Delta t}$) according to the approach of Rolstad et al. (2009):

$$\sigma_{\Delta h/\Delta t} = \sqrt{\frac{A_{cor}}{5A_{gl}}}\ \sigma_{AW} \qquad A_{gl} > A_{cor} \tag{1}$$

$$\sigma_{\Delta h/\Delta t} = \sigma_{AW} \qquad A_{gl} < A_{cor}$$





Where $A_{cor}=\pi*d_{cor}$ is the correlation area, $A_{gl}$ is the analysed glacier area and $\sigma_{AW}$ is the assessed accuracy of the elevation

change rates (explained in the next paragraph). The correlation length ($d_{cor}$) is obtained by generating semivariograms with

100000 random samples of $\Delta h/\Delta t$ values on the off-glacier areas. A binning in 30 m distance intervals and a maximum

distance of 20 km are applied. Spherical semivariogram functions are fitted to the data and an average correlation length of

387 m results from the analysed elevation change fields. Equation 1 is applied for each continuous glaciated area (icecap or

connected glaciers) and the area-weighted average of the individual ice-covered areas is taken as the region wide $\sigma_{\Delta h/\Delta t}$.

The hypsometric extrapolation of elevation change information leads to an additional uncertainty that is hard to quantify. We

employed the approach of Berthier et al. (2014). A scaling factor (we selected a factor of 2) is applied to $\sigma_{\Delta h/\Delta t}$ for the area

fraction with hypsometric extrapolation of $\Delta h/\Delta t$, in order to obtain the uncertainty of our region wide average elevation

change rates $\delta_{\Delta h/\Delta t}$.

### 4.3   Mass balances

The geodetic mass balances $\Delta M/\Delta t$ of the analysed regions are computed according to Fountain et al. (1997) by multiplying

the integrated elevation change rates by the average ice density (conversion factor). We followed the suggestion of Huss

(2013) for alpine glaciers and applied an average ice density ($\rho$) of 850±60 kg m⁻³. As revealed by Huss (2013), the

conversion factor can vary strongly. Accordingly to his findings, we applied a higher uncertainty of ±300 kg m⁻³ for the

conversion factor for observation intervals shorter than 10 years and mass budgets lower than ±200 kg m⁻² a⁻¹ (Braun et al.,

2019). Mass budgets are computed for different time intervals of all three subregions R1-3 and of single glacier basins with

at least 50% coverage by $\Delta h/\Delta t$ measurements.

In order to estimate the accuracy of the geodetic mass balances, the following error contributions are considered:

- accuracy of the elevation change rates $\delta_{\Delta h/\Delta t}$

- accuracy of the glacier outlines $\delta_S$

- uncertainty of the applied average ice density $\delta_\rho$

- potential bias due to different SAR signal penetration $V_{pen}/\Delta t$

This leads to the following formula to calculate the error of $\Delta M/\Delta t$:

$$\delta_{\Delta M/\Delta t} = \sqrt{\left(\frac{\Delta M}{\Delta t}\right)^2\left(\left(\frac{\delta_{\Delta h/\Delta t}}{\frac{\Delta h}{\Delta t}}\right)^2 + \left(\frac{\delta_S}{S}\right)^2 + \left(\frac{\delta_\rho}{\rho}\right)^2\right) + \left(\frac{V_{pen}}{\Delta t}*\rho\right)^2} \qquad (2)$$

The assessments of $\delta_S$ and $\delta_{\Delta h/\Delta t}$ are depicted in Section 5 and 6. The uncertainty contribution by potentially different SAR

signal penetration in the glacier surfaces is evaluated following the approach of Malz et al. (2018). No difference in the SAR

signal penetration in the ablation areas below the equilibrium line altitude (ELA) is assumed. These areas experience melt

throughout the year in the Tropical Andes (Kaser, 2001) and differences in the radar signal penetration in wet glacier surfaces

are small (Casey et al., 2016; Rossi et al., 2016; Ulaby et al., 1984). A linear increase of the penetration depth bias towards 5

m between the ELA and the highest peaks is employed in the accumulation areas to calculate $V_{pen}$. The late summer snow





line altitude (SLA) derived from optical satellite images is a good proxy for the ELA in the Tropical Andes (Rabatel et al.,
2012). Thus, the average ELA in the study period for the subregions is estimated based on published ELA and SLA values.
Table S3 provides an overview of the considered ELA and SLA values. Uncertainties in the estimated ELA positions do not
influence the observed mass balances. Only the penetration depth bias estimation and thus the error budget is affected. A
sensitivity analysis of an ELA mismatch on the error budget is provided in Braun et al. (2019). Specific mass balances are
calculated according to the UNESCO Glossary of Glacier Mass Balance (Cogley et al., 2011) by applying the average glacier
area of each observation period (mean glacier extent with slopes <50°).

# 5    Results

## 5.1    Glacier inventory

The glacier outlines from 2000, 2013 and 2016 of some exemplary mountain ranges from all 3 subregions are shown in
Figure 3. A clear recession of the glaciers in all subregions is obvious. The measured extents of the glaciers for all subregions
and time steps are presented in Table 1. Since cloud and snow cover do not allow for a complete coverage of subregion R1
and R2 in 2013 and no suitable SAR data covers the debris-covered ice areas in the Cordillera Blanca in 2013, a small
fraction of the glacier area (subregion R1: 2.1%; subregion R2: 4.4%) is mapped using imagery from 2014. For subregion
R1, a nearly complete mapping (99.5%) of the glacier areas in 2014 is possible. The obtained average area change rate
between 2013 and 2014 at glaciers delimited in both years is employed to correct the area change measurement in subregion
R1 for 2013 at glaciers only delimited in 2014 and vice versa. In subregion R2, the change rate between 2013 and 2016 is
applied to perform a similar correction of the area change measurements in 2013. In total, ice covered areas of 1916.6±48.3
275    km in 2000, 1571.9±43.1 km² in 2013 and 1368.1±45.5 km² are estimated for Peru. Based on the catchment divides of the
RGI6.0, a reduction in the number of glaciers from 1973 in 2000 to 1803 in 2016 is observed. The variations of the glacier
count and area in the subregions and of the whole country are summarized in Table 1 and 2. The revealed area changes of
individual glaciers in the different subregions and study periods are correlated with topographic parameters (glacier area,
median elevation, mean aspect) and plotted in Figure 4 and S5-S12.

## 5.2    Elevation changes

The obtained elevation change rates on ice-covered areas at some exemplary mountain ranges are illustrated exemplary in
Figure 5 and 6 for the periods 2000-2013 and 2013-2016, respectively. In the interval 2000-2013, a clear thinning of the
lower glacier parts in the Cordillera Vilcanota and Apolobamba is observed. The glaciers in the Cordillera Blanca and
Huayhuas show a more balanced pattern and Coropuna experienced thinning throughout most of its ice cap. An increase in
the surface lowering rates at most mountain ranges is obvious in the interval 2013-2016. Only the ice caps in subregion R3
show reduced lowering rates. Since the TDX data in 2013 does not cover the whole glacier area in subregion R3, the voids
are filled with TDX data from 2014. The glacier area covered by $\Delta h/\Delta t$ measurements using TDX data from 2014 amounts to



only 1.4% (glacier outlines from 2000). The impact of this void filling is considered negligible since the analysis uses change rates. The average measured and extrapolated surface elevation change rates of all subregions for different observation periods are listed in Table 2. The fraction of glacier area covered by $\Delta h/\Delta t$ measurements is lowest in subregion R1, when

using the SRTM DEM as a reference (Table 2). Large data voids in the SRTM data lead to a partial coverage of only 46% of the ice covered area in subregion R1 in 2000 (61% in subregion R2, 89% in subregion R3). The amount of measurements on the glaciated areas clearly increases when deriving the elevation change solely from TDX datasets (Table 2). In the period 2013-2016, a coverage of up to 80%, 69% and 89% is obtained in subregion R1, R2 and R3, respectively. Since the differential InSAR approach is used to generate the TDX DEMs, the proportion of areas affected by phase-jumps in the

unwrapped interferogram is small (~1% of the total glacier area). The major area affected by phase-jumps (at the Cordillera Vilcanota in 2016) is highlighted in Figure 6. Figures 5 and 6 indicate that the elevation change rates are altitude dependent. The hypsometric distributions of the measured elevation change rates and the respective ice covered areas are plotted in Figure 7, S13 and S14 for the each subregion in the interval 2000-2016. Surface lowering is found on areas below 5700m in subregion R1, 5800m in subregion R2 and 5900 m in subregion R3. Considerable positive elevation changes are observed in

the higher reaches of the glaciers in subregion R1.

## 5.3     Mass balances

The geodetic mass balances derived from the elevation change information of the individual subregions are listed in Table 2. In total, a mass loss of -9.18±1.10 Gt is found for Peru in the period 2000-2016. The mass loss rates show an increase between the observation periods 2000-2013 and 2013-2016. Only in subregion R3 is a slight reduction of the mass loss observed. The most prominent increase in glacier wastage is revealed in subregion R1. The mass budget changed from nearly

stable conditions of 68±102 kg m$^{-2}$ a$^{-1}$ in 2000-2013 to a specific mass loss rate of -990±476 kg m$^{-2}$ a$^{-1}$ in 2013-2016. The computed mass balances of individual glaciers in the different subregions and study periods are correlated with topographic parameters (glacier area, median elevation, mean aspect) and plotted in Figure 8 and S15-S19.

## 6     Discussion

### 6.1     Glacier retreat

We observed a dramatic recession of the glaciers throughout Peru of -29% (-548.5±65.7 km²; -1.8 % a$^{-1}$) between 2000 and 2016. Our total mapped glacier extent of 1368.1±44.5 km² in 2016 is comparable to the reported coverage of 1298.6 km² by

the recent Peruvian Glacier Inventory (UGRH, 2014), considering that UGRH did not include the glacier areas of the Bolivian Cordillera Apolobamba (~70 km²). The glacier area mapped in the RGI6.0 amounts to 1602.96 km², which is in the range of our measurements for 2000 and 2013. A direct comparison is complex, since the RGI6.0 is a blended product with outlines discriminated between 2000 and 2009 in Peru. However, visual inspection of our outlines and the RGI6.0 revealed that numerous small glaciers (especially in the southern section of subregion R1 and throughout the whole subregion R3)





mapped in the RGI6.0 are artefacts, which are caused most probably by temporal snow cover. This issue has already been
mentioned in the RGI Technical Report (RGI Consortium, 2017). Thus, our glacier inventory of Peru in 2000 has 350 less
features than the RGI6.0. By contrast, the Peruvian Glacier Inventory (UGRH, 2014) lists 2679 glaciers. This difference can
be explained by the different basin delineations applied.

The comparison of our area measurement of 1916.6±48.3 km² in 2000 and the 1st Peruvian Glacier Inventory (Hidrandina
SA, 1989) in 1970 (2041.85 km²) results in a retreat of -7% (-139.9 km²; 0.2% a$^{-1}$). However, the retreat rate is most likely
higher since not all glaciated Cordilleras were completely mapped in the 1st Peruvian Glacier Inventory (UGRH, 2014).

Figure 9 illustrates the temporal evolution of the glacier area changes and highlights that Peru's glaciers have experienced
long-term shrinkage since 1970 with strongly increasing rates in recent years. This observed trend is in accordance with the
findings of previous studies at individual mountain ranges. Silverio and Jaquet (2017) summarized area measurements of
several studies in the Cordillera Blanca (subregion R1) and revealed an increase in the loss rate from about -5 km² a$^{-1}$
between 1971 and 1996 towards -23 km² a$^{-1}$ between 2015 and 2016. Burns and Nolin (2014) reported a ~3.5 times higher
area loss rate in 2004-2010 compared to 1970-2003 in the Cordillera Blanca. Area loss rates of ~1.8 % a$^{-1}$ (>50%) since 1975
are reported by the Instituto Geofísico del Peru (2010) and López-Moreno et al. (2014) for the Cordillera Huaytapallana,
which is higher than our finding of -1.1 % a$^{-1}$ in subregion R1 in the period 2000-2013. However, their study period includes
strong El Niño events in the 1980/90s (Figure 9), which typically lead to increased glacier melt in the Tropical Andes
(Wagnon et al., 2001). Retreat measurements in subregion R2 are carried out at the Cordillera Vilcanota (Hanshaw and
Bookhagen, 2014; Salzmann et al., 2013; Veettil and Souza, 2017), the Cordillera Apolobamba (Cook et al., 2016; Veettil et
al., 2017a), the Cordilleras Carabaya, Urubamba and Vilcabamba (Veettil et al., 2017d) and at glaciers draining into the
Vilcanota-Urubamba basin (Drenkhan et al., 2018). All studies revealed significant glacier shrinkage since the 1980s with
retreat rates in the range of -0.9 to -1.7 % a$^{-1}$. These findings are comparable with our observed rate of -1.3 % a$^{-1}$ in
subregion R2 in the period 2000-2013. Drenkhan et al. (2018) revealed reduced shrinkage rates for 2010-2016 as compared
to 2004-2014. This is contradictory to our observed strong increase in glacier recession after 2013. They analysed only
glaciers in the Vilcanota-Urubamba basin, of which most are south facing. We measured the highest retreat for glaciers
facing north in subregion R2 in the period 2013-2016 (Figure S21) and only low retreat for south facing glaciers. Thus,
different spatial extents and representativeness of topographic attributes in the analysed regions leads to the mismatch of the
observed retreat trends. Moreover, this suggests that the representativeness of topographic settings needs to be considered
when doing region wide upscaling of sampled glacier change measurements. At the ice cap of the Coropuna volcano
(subregion R3), various ice coverage estimates are available going back to 1955 (Peduzzi et al., 2010; Racoviteanu et al.,
2007; Silverio and Jaquet, 2012; Ubeda Palenque, 2011; Veettil et al., 2016). The average loss rate of ~1.5 % a$^{-1}$ (1955-2015)
is lower than our estimated rate of -2.3 % a$^{-1}$ for subregion R3 in the period 2000-2013. Besides differences in the study
periods, we attribute this deviation to the fact that our estimate for subregion R3 includes numerous small glaciers located at
lower altitudes (Figure S9). These glaciers are in general more sensitive to climate change (Francou Bernard et al., 2003;



Vuille et al., 2008) in comparison to the more elevated glaciers of the Coropuna volcano. Moreover, Veettil et al. (2016) discovered an increased retreat and uplift of the SLA after ~2000 at the Corona's ice cap, supporting our findings.

For the period 2013-2016, we discovered a four times higher countrywide retreat rate compared to the period 2000-2013. The strongest increase is found in subregion R2, whereas in subregion R3 (only ~5% of total glacier area) the glacier area remained quite stable after 2013. The increased retreat rates in subregions R1 and R2 after 2013 can be attributed to the strong ENSO activities in the years 2015/16. An average ONI of 0.42 is reported for 2013-2016 and the maximum ONI of 2.6 in December 2015 indicates distinct El Niño conditions. On the other hand, an average ONI of -0.17 is revealed for

2000-2013, indicating that La Niña conditions dominated this period. Since El Niño periods typically lead to increased glacier wastage in the Tropical Andes (Vuille et al., 2008; Wagnon et al., 2001) (Section 3), our observed increased shrinkage after 2013 can be attributed to the ENSO activities in this period. The stagnation of the glacier retreat in subregion R3 in the interval 2013-2016 is difficult to explain. During the strong El Niño in 1997/98, increased precipitation was observed at the Coropuna volcano in subregion R3 (Herreros et al., 2009; Silverio and Jaquet, 2012), whereas, El Niño usually leads to

reduced precipitation rates. Veettil et al. (2016) reported positive precipitation anomalies after 2011 and clearly negative anomalies in the period 2009-2011 for the Coropuna volcano. However, the total precipitation data from ERA-Interim indicates lower precipitation rates after 2013 (Figure S23) and do not clearly indicate an increase in precipitation during El Niño 1997/98. Since the small glacier areas in subregion R3 cover mainly volcano peaks, the revealed precipitation values from the spatially coarse ERA-Interim data does not necessarily reflect the local precipitation pattern at the prominent, high

altitude volcano peaks. Moreover, the mean glacier altitude in 2013 shifted ~100 m above the ELA (Figure S24), whereas the mean glacier altitude in 2000 was nearly similar to the ELA (Figure S14). Thus, we suppose that the steady glacier conditions after 2013 in subregion R3 are caused by the allocation of the remaining ice at higher altitudes and increased precipitation, even though a strong El Niño event occurred in this period.

  The analysis of area fluctuations of individual glaciers revealed in all three subregions indicates higher recession for glaciers

with lower median elevation and for small glaciers (Figure 4 and S5-S12). This is in accordance with findings reported in previous studies (e.g. Kaser and Osmaston, 2002; Mark and Seltzer, 2005; Ramirez et al., 2001). Small glaciers have in general a more narrow altitude range as compared to larger glaciers, which can maintain the ELA below the maximum glacier elevation. A rise in SLA (a proxy of the ELA in the Tropical Andes, Section 8) is observed throughout the Peruvian Andes by various studies (Hanshaw and Bookhagen, 2014; López-Moreno et al., 2014; McFadden et al., 2011; Veettil et al.,

2016, 2017d, 2017c). This corresponds to our observed retreat pattern. Figure 4 and S5-S12 suggest that glaciers with slopes facing on average in the south/south-west direction experienced in general higher relative retreats. The total amount of lost glacier area repeats this general pattern (Figure S20-22). This can be attributed to the fact that more low lying, small glaciers with mean aspects facing southwards still exist (Figure 4 and S5-S12). Higher retreat rates in general were observed for north-orientated glaciers before 2000 (Veettil et al., 2017a; Veettil, 2018), leading already to the disappearance of small

north-facing glaciers before the start of our observation periods.





In total, we discovered that 177 glaciers disappeared in our observation period, of which most disappeared after 2013 and were south facing (Table 1 and Figure S10-S12). At the Artesonraju Glacier in the Cordillera Blanca (subregion R1), Vuille et al. (2018) projected an uplift of the ELA by 300-700 m until 2100 based on the CMIP 5 scenarios RCP 4.5 and 8.5. Thus, the proceeding climate change will lead to the further disappearance of numerous small low-lying glaciers in the Tropical

Andes within the next decades, as predicted by Ramirez et al. (2001) and Huh et al. (2017).

The gain and formation of proglacial lakes are consequences of glacier recession (Cook and Quincey, 2015), and increases the GLOF imminence of downstream areas. Veettil et al. (2017a) discovered an increase in the number of glacial lakes from 697 to 903 in the Cordillera Apolobamba and Carabaya between 1985 and 2015. In the Cordillera Vilcanota, Hanshaw and Bookhagen (2014) observed stable or increasing extents in 77% of the lakes connected to glacial watersheds. Colonia et al.

(2017) compiled an inventory of 201 potential future glacier lakes based on modelled glacier bed overdeepenings. Considering the revealed finding and reported values, a further region wide monitoring of glacier retreat and lake development is highly advisable to identify potential GLOF risks in the Tropical Andes, as also suggested by Cook et al. (2016).

## 6.2  Surface elevation changes and mass balances

The average countrywide glacier surface elevation change between 2000 and 2016 amounts to -0.359±0.068 m/a, which

corresponds to a mass budget of -357±43 kg m$^{-2}$ a$^{-1}$. Extremest surface lowering is revealed for subregion R1 in the period 2013-2016 (Table 2). However, the stable surface elevations before 2013 in subregion R1 suppress the long-term average value. The nearly balanced budget contradicts the observed glacier retreat of -15% in subregion R1 in the interval 2000-2013 at first glance. The mean surface elevation change in the retreat areas amounts to -0.28 m/a, clearly indicating mass loss in the deglaciated areas. However, elevation gain is found at high altitudes. The more La Niña-like conditions of the ENSO in

the period 2000-2013 (Section 10) and an increase in precipitation in this region due to stronger upper-tropospheric easterlies (Schauwecker et al., 2017) has most probably led to higher accumulation rates. ERA-Interim reanalysis data also shows an increase in total precipitation in this period, especially around 2007 (La Niña event). Thus, the accumulation gain in the upper reaches balanced the ice losses at the termini, even though temperatures increased (Schauwecker et al., 2017). In all subregions an increase in temperature is found in the reanalysis data for 2000-2013, but the strongest positive precipitation

anomaly is found in subregion R1 (Figure S23 and S25). This explains the mass losses in subregions R2 and R3 in this period.

Skin temperature was still above the long-term average in subregion R1 and R2 after 2013 (Figure S25). The downward surface thermal radiation shows an increase in subregion R1 (Figure S26), whereas total precipitation decreased in subregion R2 and remained nearly stable in subregion R1 (Figure S23). These climatic settings explain the more negative mass

balances in both subregions in the period 2013-2016, which also correlate with the strong El Niño activities in this interval (Figure 9). Only at subregion R3 did the thinning rate reduce after 2013, although El Niño conditions dominated. We attribute this, like the stable glacier area, to higher precipitation rates at the ice capped volcanos and the allocation of the





remaining ice masses at high elevations (Section 10 for more details). Moreover, the revealed spatial pattern of mass balances after 2013, with highest mass loss rates at the northern most subregion R1 (Table 2), matches the observed trend of
higher river runoff towards northern Peru during strong El Niño events (Casimiro et al., 2012, 2013).

The analysis of the mass balance of individual glaciers and topographic parameters in subregions R2 and R3 reveals a trend towards higher mass losses for glaciers facing north/north-east in the period 2000-2016 (Figure 8, S15 and S16). This trend agrees with observations by Soruco et al. (2009a) in the Bolivian southern wet outer tropics. In the interval 2000-2016, no clear dependency of the specific glacier mass balances on the median elevation or aspect is obvious in subregion R1 (Figure
8). However, Figure 6 suggests a trend towards higher surface lowering rates on the western slopes of the Cordillera Blanca in the period 2013-2016. The subregion-wide analysis of glacier elevation changes in dependency to aspect does not reveal any clear trend (Figure S27), however, when analysing only the Cordillera Blanca an east to west gradient is obvious (Figure S28). We attribute this to the changed precipitation pattern during El Niño. Higher precipitation rates are typically found on east-facing slopes at the Cordillera Blanca, fed by moist air from the Amazon basin (Garreaud et al., 2009). However, during
El Niño the westward flow of the moist air is hampered by stronger westerlies (Vuille, 2013), which increases the precipitation gradient across the Cordillera Blanca.

Long-term glaciological mass balance measurements are only available for two glaciers (WGMS, n.d.). For the Artesonraju and Yanamarey glaciers, both located in subregion R1, average mass budgets of -801 kg m$^{-2}$ a$^{-1}$ and -1.164 km/m²a (2004-2016) are revealed from field measurements, whereas we observed -333±77 kg m$^{-2}$ a$^{-1}$ and -451±77 kg m$^{-2}$ a$^{-1}$ (2000-2016),
respectively. Both methods show higher loss rates for the Yanamarey Glacier compared to the Artesonraju Glacier. The deviation of ~60% between both approaches can be partly attributed to different observation intervals. Moreover, glaciological mass balance measurements are typically based on some stake measurements in the lower, often most accessible, ablation zone and only a few measurements in the accumulation region. At the Zongo Glacier, Bolivia, Soruco et al. (2009b) performed a comparison of different mass balance methods (glaciological, hydrological and geodetic) and
revealed a strong offset in the glaciological mass balance estimates. The authors suggest that the limited number of field measurement points is not representative of the whole glacier and that the interpolation between the measurement sites is not valid to obtain glacier wide specific mass balance information. On the other hand, to obtain glacier wide mass balance estimates using the geodetic method, there is interpolation over data voids in the elevation change fields needed (in our case we used the hypsometric elevation change distribution of the subregion). Thus, we attribute the offset between the results
from both methods to uncertainties due to interpolation, different observation periods and the potential bias of the glaciological method towards higher loss rates due to the higher density of measurements in the ablation region.

In subregion R1, Huh et al. (2017) derived volume losses ranging from -0.019 km³ to -0.150 km³ at six glaciers in the Cordillera Blanca from various elevation datasets in the period 1962-2008. This ice depletion corresponds to surface lowering rates of -0.20 m/a to -1.4 m/a. Mark and Seltzer (2005) reported lowering rates ranging from -0.14 to -0.59 m/a for
three glaciers at the Nevado Queshque (Cordillera Blanca, subregion R1) in the interval 1962-1999. The findings of both





studies are in the range of our revealed elevation change rates for subregion R1 of -0.208±0.065 m/a and -1.067±0.273 m/a in the periods 2000-2016 and 2013-2016, respectively.

In subregion R2, Salzmann et al. (2013) estimated an ice volume loss in the Cordillera Vilcanota of ~40-45% between 1962 and 2006 based on ice thickness derived from glacier inventory parameters and thickness-volume scaling. The authors
pointed out that nearly all ice loss occurred after 1985, leading to a lowering rate of about -0.39 m/a for the period 1985-2006. This value is similar to our measured average surface lowering in subregion R3 of -0.440±0.069 m/a in the interval 2000-2013.

At the Coropuna volcano in subregion R3, Racoviteanu et al. (2007) measured an average glacier surface lowering of -5 m (-0.1 m/a) based on a SRTM DEM and digitizing of a topographic map from 1955. Peduzzi et al. (2010) calculated a mean
surface lowering of -0.2±0.3 m/a for the period 1955-2000/02. Our revealed average surface lowering of -0.44±0.045 m/a at the ice cap of the Coropuna volcano indicates an increased glacier wastage after 2000, which correlates with the increase in glacier retreat and SLA uplift observed by Veettil et al. (2016) since 2000.

On the countrywide scale, we observed ~19 % less mass loss than Braun et al. (2019) (Region 02-04) in the period 2000-2012/13. However, their analysis also includes the glacier areas in Bolivia, like the Cordillera Real, where significant glacier
wastage is reported (e.g. Soruco et al., 2009a), leading to higher average mass loss rates. The change rates of subregion R1 and R2 in the period 2000-2012/13 and the results of Braun et al. (2019) in Region 02 and 03 show good agreement, even though they are based on different glacier inventories. However, we computed more negative mass budgets in subregion R3 as compared to Braun et al. (2019) (Region 04). This can be partly explained by the differences in the region delineation. Braun et al. (2019) included the few ice-covered volcanoes in south-west Bolivia. However, a more considerable reason for
this offset is the fact that we found the highest amount of misclassified ice covered areas in the RGI 6.0 in this subregion (Section 9). This explains the bias toward lower mass loss rates in the results of Braun et al. (2019), who measured surface elevation changes based on the RGI 6.0. In a continent wide analysis of geodetic mass budgets, like that of Braun et al. (2019), it is beyond the scope of the studies to map glacier outlines fitting to the whole elevation database as well. However, the revealed offset suggests the inaccuracy caused by imprecise glacier outlines and highlights the need for large-scale
temporally consistent glacier outlines.

Gardner et al. (2013) carried out a comprehensive worldwide estimate of the glacier contribution to sea level rise. They computed a mass budget of -1080±360 kg m$^{-2}$ a$^{-1}$ (2003-2009) at the "low latitudes" (RGI region definition) by means of the extrapolation of glaciological measurements. Their ice loss rate is about three times higher than our countrywide average of -357±43 kg m$^{-2}$ a$^{-1}$. This offset is similar to the deviation with glaciological mass budget estimates of individual glaciers and
thus can be attributed to the same causes (see above). Moreover, the representativeness of the settings of the sampled glaciers, which is not assessed by Gardner et al. (2013), can also strongly influence region wide estimates as discussed in Section 10. A comparison with GRACE measurements by Jacob et al. (2012) is also difficult, since their spatial domain covers the whole of South America, only excluding Patagonia. Thus, the mass balance of the Peruvian glaciers cannot be




disentangled from their results. These factors depict the limitations of using present global mass balance estimates at the country or even mountain range level.

The above-mentioned global or continent-wide analyses do not cover multiple periods and thus do not provide any information on temporal variability of the mass balances. Our analysis reveals strong temporal variations in the glacier changes that correlate with changing climate conditions and specific climatic events. These findings underline that mono-temporal analysis, especially when using short time intervals, can be biased by short-term climate anomalies like El Niño and highlights the need for further monitoring of the proceeding glacier recession in the Tropical Andes.

## 7      Conclusions

The glaciers throughout Peru are strongly affected by changing climatic conditions, leading to considerable ice losses. In this comprehensive study we revealed a glacier recession of $-548.5\pm65.7$ km² (-29%) in the period 2000-2016 and negative regional mass budgets of up to $-990\pm476$ kg m$^{-2}$ a$^{-1}$ (northern wet outer tropics; 2013-2016). A strong increase in the countrywide mass and area loss rates from $-184\pm45$ kg m$^{-2}$ a$^{-1}$ and -1.4% a$^{-1}$ in the period 2000-2013 to $-836\pm188$ kg m$^{-2}$ a$^{-1}$ and -4.3 % a$^{-1}$ in the interval 2013-2016 is shown. This amplified glacier wastage can be attributed to the strong El Niño in 2015/16. Spatial and temporal differences of the change rates of the studied subregions correlate with skin temperature and total precipitation trends derived from ERA-Interim reanalysis data and reported regional climatic variations. The analysis of area changes of individual glaciers indicates that the highest relative area change rates are found for low lying and small glaciers, validating the prediction of the disappearance of numerous small glaciers located at low altitudes in the Tropical Andes.

Our results provide the first multi-temporal region wide and spatially detailed analysis covering all glaciated areas in Peru, providing fundamental data for projecting future glacier changes, water resource management schemes and further glacier monitoring. The observed changes highlight the dramatic progression of glacier recession throughout Peru, which will lead to considerable socio-economic issues in this region. The increasing GLOF risk, due to the gain and formation of glacial lakes is just one aspect. The future contribution of glacier meltwater to the regional runoff puts the continuous water availability for irrigation, mining, hydropower generation and drinking water supply, especially during dry season, at risk. Therefore, we highly advocate resuming and further extending the glacier monitoring in the Tropical Andes, not solely to gain scientific knowledge, but also to provide important information for local authorities and decision-makers regarding water resource management and civil protection.

**Author Contributions:** TS designed and led the study, processed and analysed the data and wrote the manuscript. TS, PM and CS developed jointly the analysis routines for elevation change and mass balance computations. SL performed the SAR coherence computations. AC contributed to the interpretation of the data and provided field measurements. MB initiated and supervised the project. All authors revised the manuscript.





**Competing interests:** The authors declare no conflict of interest. The founding sponsors had no role in the design of the
510 study; in the collection, analysis, or interpretation of data; in the writing of the manuscript, and in the decision to publish the
results.

**Acknowledgements:** This work was financially supported by the DLR/BMWi grant GEKKO (50EE1544), by the Deutsche
Forschungsgemeinschaft (DFG) in the framework of the priority programme "Antarctic Research with comparative
investigations in Arctic ice areas" SPP 1158 by the grant DFG BR2105/9-1 and the priority programme "Regional Sea Level
Change and Society" by the grant DFG BR2105/14-1, as well as the HGF Aliance Remote Sensing & Earth System
Dynamics. The authors would like to thank the German Aerospace Center for providing TanDEM-X and TerraSAR-X data
free of charge under AO XTI_GLAC0264 and AO ARC_HYD1763. Landsat data was kindly provided via USGS Earth
Explorer and Sentinel-1 data was provided by ESA via Copernicus Open Access Hub. SRTM data was provided by NASA
LP DAAC

**Data and materials availability:** Elevation change fields will be available via the World Data Center PANGAEA operated
by AWI Bremerhaven after acceptance of the manuscript. Glacier area information and glacier-specific results will also be
made available through submission to the World Glacier Monitoring Service and GLIMS.

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



**Figures:**

**Figure 1.** General maps of study region. Panels (a-c): glacier subregions in Peru according to Sagredo and Lowell (2012); (a) subregion R1: northern wet outer tropics; (b) subregion R2: southern wet outer tropics; (c) subregion R3: dry outer tropics. Panel (d): overview map of Peru. Coloured rectangles indicate the locations of the subregions (same frame colours). Light blue areas: glacier coverage based on RGI 6.0. Background: SRTM DEM © NASA







**Figure 2.** Off-(red) and on-glacier (light blue) area and off-glacier elevation change (blue dots) distributions in dependency on slope in subregion R1 for the period 2000-2016. Error bars represent NMAD of Δh/Δt values in the individual slope interval. Dotted line indicates the applied slope threshold (see Section 4.2). Glacier area measurements are based on the glacier outlines from 2000. Note: For better representation, on-glacier areas are scaled by a factor of 10. Plots for other subregions are provided in the Supplementary material.





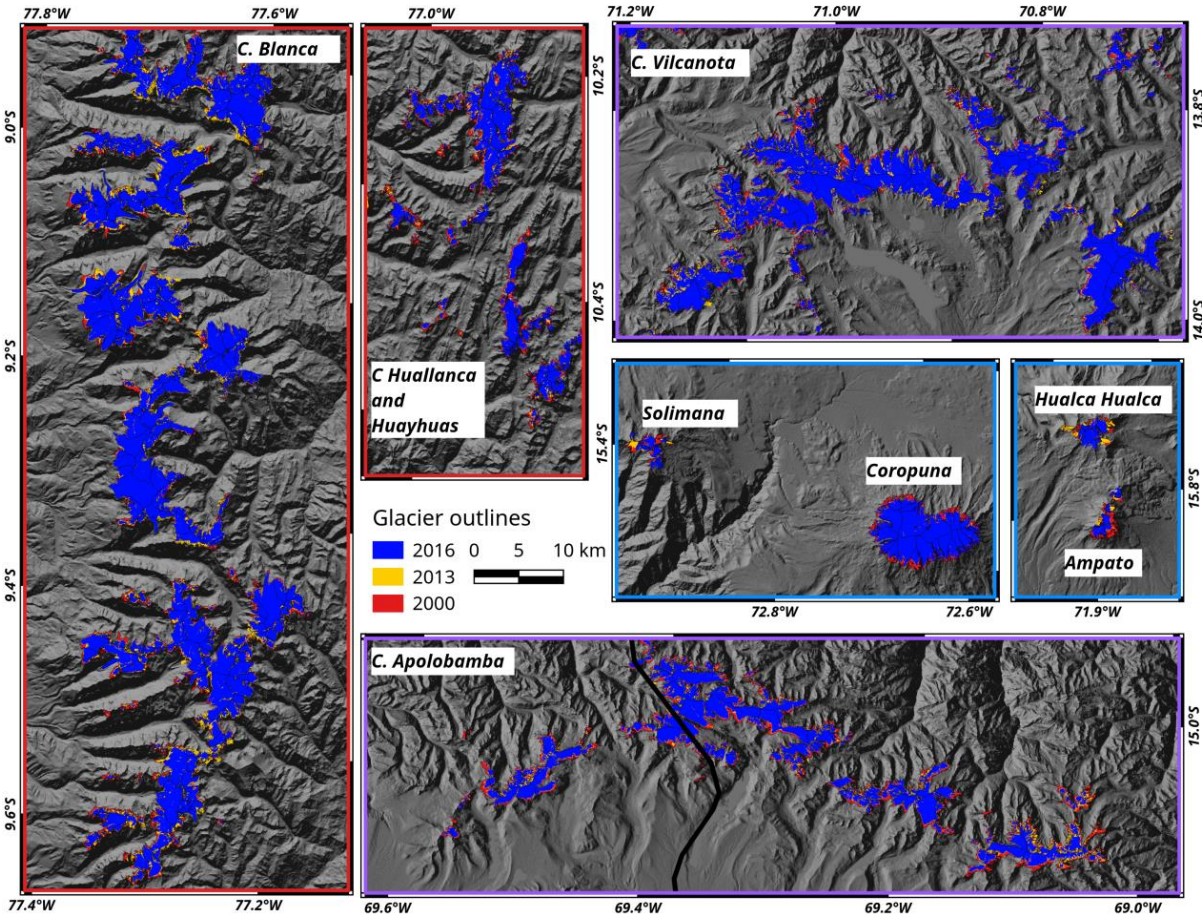

**Figure 3.** Exemplary glacier outlines at mountain ranges in all three subregions in the years 2000, 2013 and 2016. The frame colour of the panels indicates the subregions (see **Figure 1**). Background: SRTM DEM hillshade © NASA







**Figure 4.** Polar plot of relative area changes (dot colour) in subregion R1 in the period 2000-2016 of individual glaciers. Dot size: glacier size in 2000; Radius: median elevation; Orientation: mean aspect. Red circle: equilibrium line altitude (ELA), see also Table S3. Plots for other subregions and time intervals are provided in the Supplementary material.





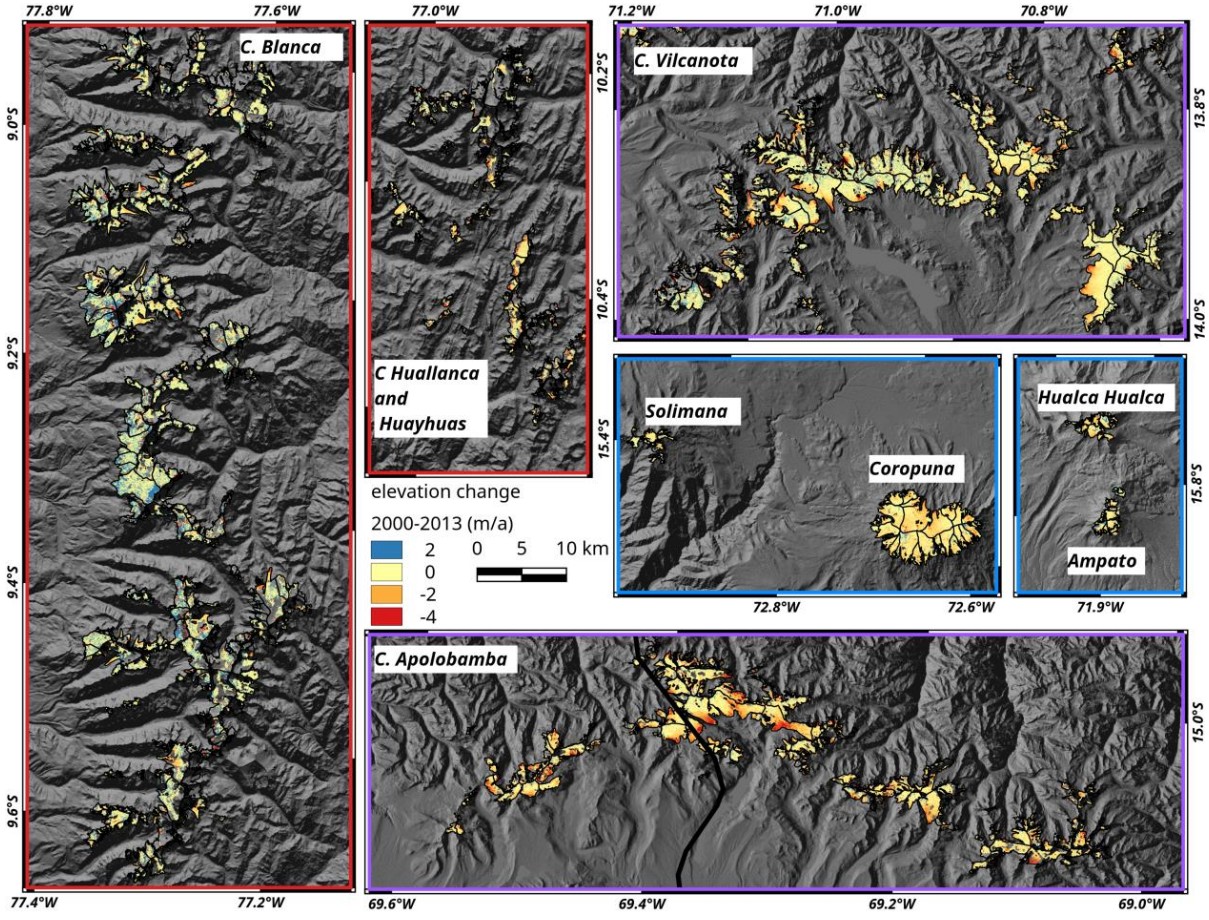

**Figure 5.** Surface elevation changes in the period 2000-2013 at mountain ranges in all three subregions. The frame colour of the panels indicates the subregions (see **Figure 1**). Background: SRTM DEM hillshade © NASA



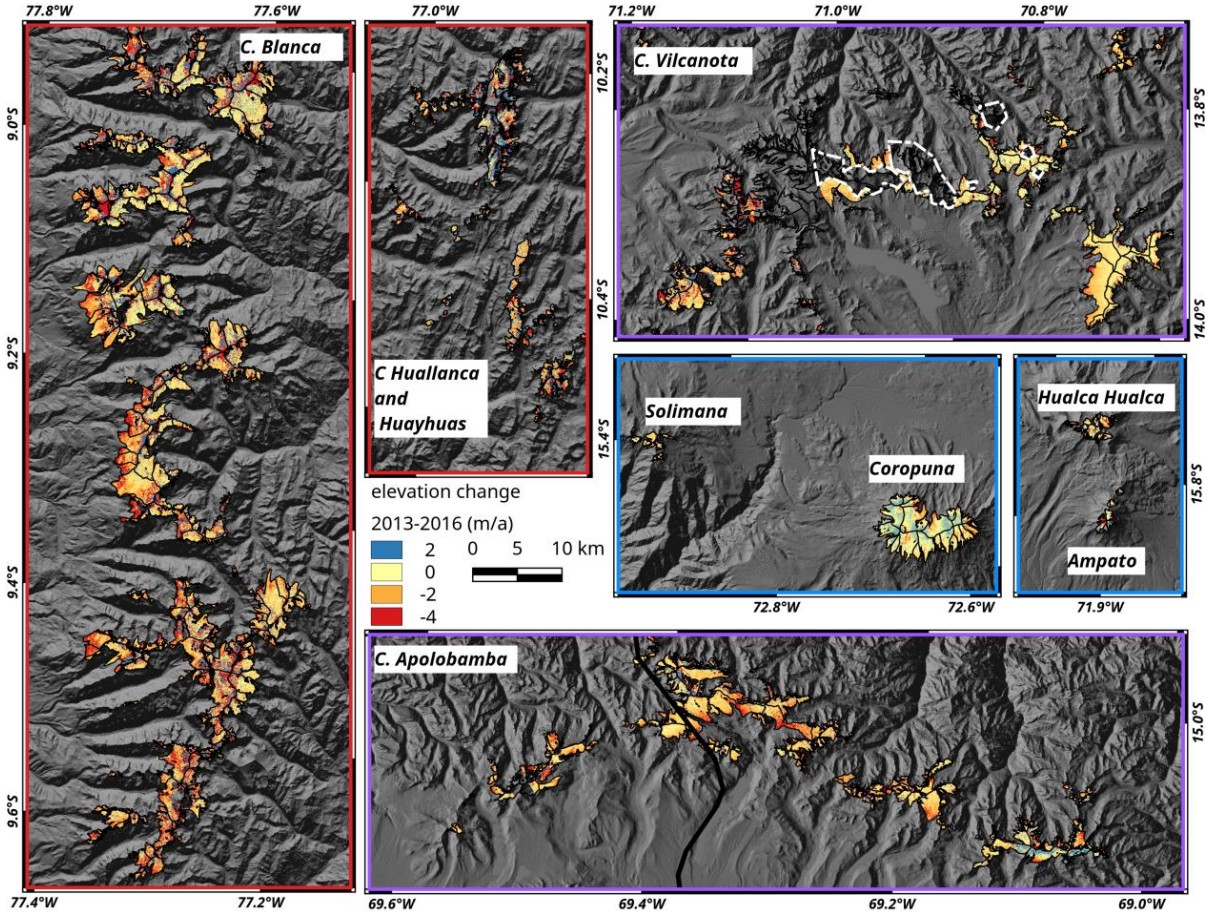

**Figure 6.** Surface elevation changes in the period 2013-2016 at mountain ranges in all three subregions. The frame colour of the panels indicates the subregions (see **Figure 1**). White dashed line: masked out areas affected by phase jumps in the unwrapped interferogram. Background: SRTM DEM hillshade © NASA





**Figure 7.** Hypsometric distribution of glacier area with elevation change (Δh/Δt) measurements (red) and total glacier area (light blue) in subregion R1 in the interval 2000-2016. Blue dots represent the mean Δh/Δt value in each elevation interval. Error bars indicate NMAD of Δh/Δt for each hypsometric bin. Grey areas mark the lower and upper 1% quantile of the glacier area distribution. Black dashed line: mean glacier elevation; Red dashed line: equilibrium line altitude (ELA), see also Table S3. Area measurements are based on the glacier outlines from 2000, considering only regions with slopes below applied slope threshold (50°, see Section 4.2). Plots for other subregions are provided in the Supplementary material.



**Figure 8.** Polar plot of specific mass balance (dot colour) of individual glaciers in subregion R1 in the period 2000-2016 of individual glaciers. Dot size: glacier size in 2000; Radius: median elevation; Orientation: mean aspect. Red circle: equilibrium line altitude (ELA), see also Table S3. Note: only glaciers with elevation change information >50% are included. Plots for other subregions and time intervals are provided in the Supplementary material.





**Figure 9.** Temporal evolution of (a) relative glacier area changes, (b) observed specific mass balance (spMB, colour shaded areas represent the uncertainty of the corresponding measurements) and (c) Oceanic Nino Index (ONI) in the period 1970-2016. *Area measurement in 1970 taken from the 1$^{st}$ Peruvian Glacier Inventory (Hidrandina SA, 1986), see also Section 6.1




## Tables:

**Table 1.** Measured glacier extents for different years and regions.

| year | $S$ (km²) | $\delta_S$ (km²) | $n$ |
|---|---|---|---|
| Subregion R1 | | | |
| 2000 | 910.1 | 32.3 | 1162 |
| 2013 | 774.9 | 27.6 | 1159 |
| 2016 | 657.0 | 26.7 | 1088 |
| Subregion R2 | | | |
| 2000 | 893.9 | 35.7 | 702 |
| 2013 | 720.8 | 33.0 | 639 |
| 2016 | 633.2 | 35.5 | 633 |
| Subregion R3 | | | |
| 2000 | 112.6 | 3.8 | 109 |
| 2013 | 76.2 | 3.2 | 104 |
| 2016 | 77.9 | 3.0 | 82 |
| All subregions (country wide) | | | |
| 2000 | 1916.6 | 48.3 | 1973 |
| 2013 | 1571.9 | 43.1 | 1903 |
| 2016 | 1368.1 | 44.5 | 1803 |

$S$: measured glacier area
$\delta_S$: uncertainty of measured glacier area
$n$: number of glacier catchments (delineations based on RGI 6.0)





**Table 2.** Measured area, surface and mass changes for different periods and regions. *mean glacier area of observation interval is used to calculate specific mass balances (see Section 4.3)

| period | $dt$ a | $dS$ km² | % | $dS/dt$ Km² a$^{-1}$ | $dh_M/dt$ m a$^{-1}$ | $dh_E/dt$ m a$^{-1}$ | $S_M$ % | $\Delta M/\Delta t$ Gt a$^{-1}$ | kg m$^{-2}$ a$^{-1}$* | $\Delta M$ Gt |
|---|---|---|---|---|---|---|---|---|---|---|
| **Subregion R1** | | | | | | | | | | |
| 2000-2012 | 11.98 | - | - | - | -0.024±0.067 | -0.017±0.104 | 46 | -0.012±0.087 | -15±111 | -0.15±1.04 |
| 2000-2013 | 12.94 | -135.2±42.5 | -15 | -10.4 | 0.055±0.61 | 0.073±0.093 | 48 | 0.053±0.008 | 68±102 | 0.68+1.03 |
| 2000-2016 | 16.65 | -253.2±41.9 | -27 | -15.8 | -0.236±0.042 | -0.208±0.065 | 47 | -0.148±0.077 | -205±107 | -2.47±1.28 |
| 2013-2016 | 3.71 | -117.9±38.4 | -15 | -39.3 | -1.208±0.228 | -1.067±0.273 | 80 | -0.649±0.312 | -990±476 | -2.41±1.16 |
| **Subregion R2** | | | | | | | | | | |
| 2000-2013 | 13.02 | -173.1±48.6 | -20 | -12.4 | -0.400±0.069 | -0.440±0.095 | 61 | -0.318±0.077 | -412±100 | -4.14±1.01 |
| 2000-2016 | 16.68 | -260.6±50.4 | -29 | -16.3 | -0.500±0.050 | -0.512±0.074 | 53 | -0.371±0.064 | -511±88 | -6.19±1.06 |
| 2013-2016 | 3.67 | -87.6±48.5 | -12 | -43.8 | -0.972±0.187 | -0.916±0.244 | 69 | -0.538±0.256 | -834±396 | -1.97±0.94 |
| **Subregion R3** | | | | | | | | | | |
| 2000-2013 | 12.93 | -36.4±5.0 | -32 | -2.6 | -0.390±0.047 | -0.394±0.052 | 89 | -0.038±0.008 | -395±85 | -0.49±0.11 |
| 2000-2016 | 16.65 | -34.7±4.9 | -31 | -2.2 | -0.322±0.037 | -0.328±0.042 | 87 | -0.031±0.007 | -328±74 | -0.52±0.11 |
| 2013-2016 | 3.71 | 1.7±4.4 | 2 | 0.9 | -0.231±0.187 | -0.210±0.208 | 89 | -0.014±0.025 | -184±329 | -0.05±0.09 |
| **All subregions (country wide)** | | | | | | | | | | |
| 2000-2013 | 12.98 | -344.7±64.8 | -18 | -24.6 | -0.220±0.064 | -0.198±0.091 | 56.9 | -0.302±0.074 | -184±45 | -4.43±1.00 |
| 2000-2016 | 16.66 | -548.5±65.7 | -29 | -34.3 | -0.370±0.045 | -0.359±0.068 | 52.2 | -0.550±0.067 | -357±43 | -9.18±1.10 |
| 2013-2016 | 3.69 | -203.8±62.0 | -13 | -101.9 | -1.049±0.208 | -0.953±0.256 | 75.6 | -1.201±0.271 | -836±188 | -3.95±0.96 |

$dt$: observation period (mean time lag between DEM dates)

$dS$: glacier area change

$dS/dt$: glacier area change rate (mean time lag between glacier inventories is used)

$dh_M/dt$: average measured surface lowering rate

$dh_E/dt$: average extrapolated surface lowering rate

$S_M$: fraction of glacier area covered by $dh_M/dt$ measurements (below slope threshold)

$\Delta M/\Delta t$: mass balance (average and specific)

$\Delta M$: Total mass change in the observation period