# Peer review of "Changes of the tropical glaciers throughout Peru between 2000 and 2016 – Mass balance and area fluctuations"

_The Cryosphere, 2018_

## Referee Comment (RC1) · Christian Huggel (Referee) · 31 Mar 2019

In my opinion this is a solid study with important new results and there is no doubt that I would like to see this paper published, eventually. While I think that the methods are good state of the art there are a few issues in the results which irritate me and make me wonder whether there are some more basic problems with data collection and analysis which I detail further below. To start I'm impressed by the amount of work done by the authors on a generally high level of data analysis, well presented. It adds new insights on glacier changes (area, surface changes and mass balance) which were previously not known on this level of detail and spatial coverage. I also like the discussion section

which is transparent, comprehensive and encompasses the (full) coverage of available literature. In this discussion section the authors critically analyze a number of differences (and similarities) between their results and those of other studies. I can follow this discussion and I think it is mostly appropriate but I'm wondering whether there are underlying errors or uncertainties in terms of data sampling or analysis that may have gone undiscovered. I list here a number of possible problem areas: The authors did not measure the full extent of the glaciers, and transparently report on it but the effect and possible uncertainties involved are not clear to me. The glacier area changes reported in Figure 9 and Table 2 contain numbers that raise some questions. An area loss of only about 5% from 1970 to 2000 is in contradiction to what is generally reported, indicating values of 15-20% (Salzmann et al. 2013, Silverio and Jaquet 2004, others, incl. unpublished data). The authors indicate some aspects about incomplete inventories, or sampling issues. I'm not sure whether this large discrepancy can be explained by the mentioned aspects but urgently needs to be clarified. I'm also irritated by the error indications related to glacier area changes reported in Table 2, of up to 30% which is much higher than what is commonly achieved in remote sensing based mapping studies (ca. up to 5%). This also needs to be clarified. I have seen many glacier mapping studies in the tropical Andes (published, or reviewed) which had errors because of inappropriately selected images with snow coverage which then resulted in erroneous glacier change results. I can't say whether this study is affected by a similar problem. In any case the authors should carefully review the literature they cite and whether some of these studies have such errors (at Coropuna for instance some published studies have such errors). I'm surprised by the drastic change of glacier and mass balance change of 2000-2013 vs 2013-2016. The authors list a number of plausible reasons, and I think the increased precipitation (accumulation) at high altitudes is a very important finding here. Nevertheless, important open questions remain. Fig. 9 indicates a change in the El Niño Index around the year 2013, changing from slightly negative to strongly positive. Reported mass balance and (high altitude) surface change can certainly be explained with this mechanism to some extent. But it

is unclear (and not plausible) how precipitation changes would immediately translate into rather drastic changes in glacier area, even if the response of tropical glaciers through feedback processes including precipitation and albedo changes is more direct than in mid-latitude glaciers. The comparison of their results with ground based mass balance measurements (glaciological method) are very significant. The authors are right that there are problems with the mass balance measurements which in fact are very challenging on these glaciers. Nevertheless, the authors should investigate this issue in more depth. I would also recommend to look in more detail on locally available field data which co-author Alejo Cochachin disposes of. The measurement interval (2000-2013) could have an effect, and changes towards more negative glacier mass balances could have been started earlier than 2013. Also, just as an additional information, according to mass balance measurements we did in collaboration with the Peruvian colleagues indicate that mass balances (since 2010) are much more negative in the Cordillera Blanca than in the Cordillera Vilcanota. All these points, open questions and uncertainties leave me with considerable doubts whether there are (basic?) problems with data collection, processing and analysis. For me these are the fundamental points that absolutely need to be clarified before this study can be published. I encourage the authors to do a serious investigation about these issues such that we can have reasonable confidence that the reported results reflect the reality and are not distorted by any errors.

Christian Huggel, University of Zurich

---

## Referee Comment (RC2) · Duncan J. Quincey (Referee) · 26 Apr 2019

This manuscript presents geodetic mass balance calculations and glacier area fluctuations for Peruvian glaciers for the period 2000-2016. The methods are robust, and the key findings are substantial – specifically that area and mass have reduced considerably over this time period, with a notable increase in the rate of loss during the latter years (2013-2016). A particular highlight is the comprehensive discussion of the study findings in the context of previous work. Despite its density, a clear path is navigable throughout and the argument is strong. The analysis is also very honest about where problems in the current work may lie. The only area where I think the authors

need to think again is in the suggestion of the strong El Niño event of 2015 as the primary reason for the rapid change in area and mass loss rates. It may be just about conceivable that changes in temperature/precipitation/humidity could impact mass balance almost immediately, but the magnitude of the change is (too) substantial for this to be the only factor, and the idea that a warm and dry event could also impact on glacier area to such a degree, within a single year, cannot hold. This requires some further investigation/consideration/analysis. Otherwise, I am very much in favour of seeing this manuscript published, and only have the following minor suggestions (by line number) to make.

10: debris-covered extents were also derived by coherence mapping according to the text?

30: 'already crossed...'

39: 'GLOF incidents...' or 'GLOF threats...'?

102: 'continuous...'

113: here and elsewhere check your cross-referencing to different sections. This one should be Section 4 (I think) – others later in the manuscript refer to sections 8, 9 and 10 that don't exist

122-123: more negative because of the lack of accumulation is what I think you mean here... but the previous sentence that refers to reduced ablation is contradictory to a more negative mass balance, so this needs clarification

173: use the correct GLIMS reference that comes with the download...

266: 'example' not 'exemplary'

275: missing power on first km

280: use of exemplary twice (though it should again be 'example' I think

315: 'temporary' not 'temporal'

349: Coropuna?

386-393: though interesting, this paragraph is only partially relevant here and could probably be cut

395: 'The most extreme surface lowering. . .'

Figure 3: caption should read 'example' not 'exemplary', but moreover I'm not sure what the value of the figure is since we can see most of this in Figure 1?

Figure 7: this caption needs some work I think. It took an age to work out that the red bars were vs the blue bars. How about 'Hypsometric distribution of measured glacier area with elevation (red) and total glacier area with elevation (blue), with mean dh/dt values in each elevation interval (blue dots). . .'?

---

## Author Response (AR1)

Response to the editor comments on "Changes of the tropical glaciers throughout Peru between 2000 and 2016 – Mass balance and area fluctuations" by Thorsten Seehaus et al.

Francesca Pellicciotti (Editor)

Received: 04 July 2019

**First of all we want to thank the Francesca Pellicciotti for editing the manuscript and constructive comments. All comments have been taken into account and a list of answers and undertaken actions is given below. Answers are in blue font color**.

In the Introduction, when the authors discuss the existing quantification of elevation changes and mass balance or ice volume changes you should mention the work by Mernild et al (2017), as they calculate surface mass balance for the entire Andes cordillera (using a mass balance model) even if they do not report them specifically. I would be curious to know how their calculations compare with yours (even if of course their period of simulations is much longer, but there are almost ten years of overlap)?

Thank you very much for pointing us to this publication. We considered it in the Introduction and the Discussion:

"Large-scale mass balance estimates covering Peru are the following: a mass balance estimation for the "low latitudes" of -1080±360 kg m-2 a-1 based on the upscaling of glaciological mass balance measurements covering the period 2003-2009 (Gardner et al., 2013); modelled surface mass balance of -1550±620 kg m-2 a-1 for the Andes north of 27°S in the period 1979-2014 (Mernild et al. 2017), a upscaled mass balance of -2 ± 2 Gt a-1 (-1030 ± 830 kg m² a-1) for the same region using glaciological and geodetic mass balances between 2006 and 2016 (Zemp et al. 2019), a mass balance calculation throughout South America (excluding Patagonia) of -6±12 Gt a-1 using space borne gravimetric measurements from the Gravity Recovery and Climate Experiment (GRACE) for the period 2000-2010 (Jacob et al., 2012); and a geodetic mass budget of -0.49±0.09 Gt a-1 (-227±42 kg m-2 a-1, ice density scenario: 850 kg m-3) derived from InSAR measurements for the period 2000-2012/13 including glaciers in Bolivia (Braun et al., 2019)."

"The continent-wide surface mass balance simulation by Mernild et al. 2017 revealed an average mass change of -1230±690 kg m² a-1 for the Andes north of 27°S in the epoch 2000-2014, which is much higher than our country-wide average of -169±43 kg m² a-1 for the period 2000-2013. This large offset can only be partly explained by the different study domains, time intervals and glacier inventories applied, and could be cause by limitations in the applied statistical downscaling of global circulation data."

Minor:
_Data: it should be plural, despite large use as singular, especially in American English. I would suggest correcting that throughout.

corrected

_Glaciated: I would replace with glacierised.

corrected

_Line 99: would you have some more references for the effect of La Nina and El Nino? This seems an important effect after all.

We added 2 more references (Garreaud et al., 2009; Maussion et al., 2015)

_Line 31 page 1: it is known to cause socio-economic issues: either provide one or more references for this (to justify the "It is known to") or change into will likely cause…

We added citation of a recently published paper regarding this issue: (Drenkhan et al. 2019)

_Please check the correctness of all your references in the Bibliography. I think some are needs some proof, e.g. Silverio and Jaquet, a capital A is needed.

We checked the Bibliography.

_Line 81, provide the standard deviation or coefficient of variation of the annual temperature, to back your statement of almost no seasonality

This value is based on Sagredo and Lowell (2012) (cited just above). The do not provide information regarding variation. Thus, we tried to infer the seasonality (annual variability from their graphs). We added the following statement:

"variability of mean monthly temperature ~1 °C"

_Line 83 and related to the above: what is an annual seasonality of 4°C? I mean, what is the 4°C and how is the seasonality defined/calculated?

We replaced the word seasonality by "annual variability of the mean monthly temperature". We hope to be more clear now.

_Sometimes you use Therefore in a bit inappropriate way, e.g. line 206. It could just be removed. Please check through the manuscript.

Done, and some "Therefore" are removed.

_Line 108: change DEMS are applied into are differentiated, or used to calculate…. Applying a DEM is a weird expression.

Corrected.

Response to the interactive comment on "Changes of the tropical glaciers throughout Peru between 2000 and 2016 – Mass balance and area fluctuations" by Thorsten Seehaus et al.

Duncan J. Quincey (Referee)

**First of all we want to thank the reviewer for constructive comments on our manuscript. All comments have been taken into account and a list of answers and undertaken actions is given below. Answers are in blue font color.**

This manuscript presents geodetic mass balance calculations and glacier area fluctuations for Peruvian glaciers for the period 2000-2016. The methods are robust, and the key findings are substantial – specifically that area and mass have reduced considerably over this time period, with a notable increase in the rate of loss during the latter years (2013-2016). A particular highlight is the comprehensive discussion of the study findings in the context of previous work. Despite its density, a clear path is navigable throughout and the argument is strong. The analysis is also very honest about where problems in the current work may lie.

The only area where I think the authors need to think again is in the suggestion of the strong El Niño event of 2015 as the primary reason for the rapid change in area and mass loss rates. It may be just about conceivable that changes in temperature/precipitation/humidity could impact mass balance almost immediately, but the magnitude of the change is (too) substantial for this to be the only factor, and the idea that a warm and dry event could also impact on glacier area to such a degree, within a single year, cannot hold. This requires some further investigation/consideration/analysis.

We appreciate this comment and carried out further analysis regarding this issue.

First, we updated and advance the mass balance computation. Instead of using a constant density for volume to mass convestion, we added a 2$^{nd}$ scenario using different densities for regions below and above the ELA (e.g. Kääb et al. 2012). This scenario is more likely for glaciers in the tropics and the regional mass balance discussion is based on it (statement added in section 6.2):

"The 2nd density scenario leads to 8-21% lower country wide mass balances. As pointed out by Kääb et al. (2012), this scenario is suitable for glacier with no dh/dt due to ice dynamics and when dh/dt is clearly driven by melt or increased accumulation. Since, these conditions are typical for the Peruvian glaciers (see Section 1 and further down), we used the results of 2nd density scenario for further discussion and analyses."

This scenario lead to a less, but still pronounced change in the mass budget for both observation periods (~15-25% less).

Regarding the mass balances: The comparison of the ONI values and glaciological measurements (specific mass balances and ELA) support the suggested link between increased mass loss and El Nino and the direct (immediately) impact on of ENSO on the mass balances. Moreover, the enhanced ablation during El Niño is on the one hand side force by the changes in e.g. temperature and precipitation, but on the other side also enhance through feedback processes. For example leads the higher temperature to liquid precipitation in the ablation region, which further increases the melting. Moreover during El Niño, precipitation is reduced but also the dry season is prolonged

(delayed start of the wet season), leading to enhanced melt due to lower glacier albedo, since the fresh snow cover is missing. We added the following statements in section 6.2 and Figure S30 in the supplement.:

"Those climatic variations enhance the ablation and facilitate a positive feedback that further increase the glacier melt. The higher temperature as well as reduced and delayed precipitation, that are typical during El Niño, lead to liquid precipitation in the ablation regions and a reduced glacier albedo, enhancing the short-wave radiation absorption (e.g. Vuille et al. 2018, Maussion et al. 2015). Thus, the climatic variations explain the more negative mass balances in both subregions in the period 2013-2016, which also correlate with the strong El Niño activities in this interval (Figure 8)."

"The correlation of the annual glaciological measurements with the average ONI of the respective observation periods indicates a trend towards increased mass loss and higher ELA during El Niño conditions (Figure S30). These tendencies fit to the observations by Silvero and Jaquet (2017) and Morizawa et al. (2013) (see Section 6.1) and support our revealed correlation between the increased glacier wastage in the period 2013-2016 and the strong El Niño event during this period. "

We also added a statement (section 3), that the seasonal offset for intervals ending in 2016 (TDX coverage in ~Oct. Nov.) leads to a small bias towards more negative mass budgets.

"Typically negligible accumulation occurs during the dry season and ablation is dominating the glacier mass budget (Favier et al., 2004; Kaser, 2001; Veettil et al., 2017b). Thus, the mass balances for observation periods ending in 2016 are slightly biased towards more negative values due to this seasonal offset in the data."

Regarding the impact on glacier area: Silvero and Jaquet (2017) as well as Morizawa et al. (2013) showed that ENSO as a clear impact on the glacier area changes. Both studies report clearly increased retreat during El Niño epochs and even area gain during La Niña. E.g. Silvero and Jaquet discovered a very high retreat rate of -23 km/a in Cordillera Blanca between 2014 and 2016 (average ONI 1.1, ~5%/a area loss, we discovered also 5%/a loss at subregion R1 for the period 2013-2016) and area gain of 5.24 km² (average ONI -0.20) between 1997 and 2002. Moreover, they inferred a linear relation between area changes and ONI with an R²=0.8. Additinally the glacier in the tropical Andes are in average the thinnest worldwide (Farinotti et al. 2019). Thus, increased ablation will cause more pronounced area reduction as compared to other regions. Thus, we conclude that the El Niño conditions and the associated increased ablation in the period 2013-2016 can be attributed as the driver for the observed increased recession. The following statement was added regarding this issue in Section 6.1:

"This pattern fits also to the finding by Morizawa et al. (2013) (at Condoriri Glacier, Bolivia) and Silvero and Jaquet (2017) (at Cordillera Blanca, subregion R1). Both studies reported enhanced recession during El Niño events and even area gains during La Niña epochs. The latter study also discovered a linear relation between glacier retreat and ONI (R²=0.8) and reports an change rate of -5% a[-1] for 2014-2016 that is equal to our change rate at subregion R1 for 2013-2016. Moreover, the glaciers in the Tropical Andes are in average the thinnest worldwide (Farinotti et al. 2019). Therefore, the increased melt will lead to more pronounced changes in glacier area as compared to other glacier region. "

Otherwise, I am very much in favour of seeing this manuscript published, and only have the following minor suggestions (by line number) to make.

10: debris-covered extents were also derived by coherence mapping according to the text?

Thank you for this advise, we added a the following statement: "The mapping of debris-covered glacier extents is supported by SAR-coherence information."

30: 'already crossed. . .'

Corrected

39: 'GLOF incidents. . .' or 'GLOF threats. . .'?

Corrected

102: 'continuous. . .'

Corrected

113: here and elsewhere check your cross-referencing to different sections. This one should be Section 4 (I think) – others later in the manuscript refer to sections 8, 9 and 10 that don't exist

Thank you very much for this advice. The automatic cross-references were somehow mixed up. We corrected this problem.

122-123: more negative because of the lack of accumulation is what I think you mean here. . . but the previous sentence that refers to reduced ablation is contradictory to a more negative mass balance, so this needs clarification

We appreciate the reviewer's comment. The review is right. We rephrased this paragraph in order to be more clear:

"Typically negligible accumulation occurs during the dry season and ablation is dominating the glacier mass budget (Favier et al., 2004; Kaser, 2001; Veettil et al., 2017b). Thus, the mass balances for observation periods ending in 2016 are slightly biased towards more negative values due to this seasonal offset in the data. "

173: use the correct GLIMS reference that comes with the download. . .

Thank you for this advice. We changed the reference accordingly.

266: 'example' not 'exemplary'

Corrected

275: missing power on first km

Corrected

280: use of exemplary twice (though it should again be 'example' I think)

Corrected

315: 'temporary' not 'temporal'

Corrected

349: Coropuna?

Corrected

386-393: though interesting, this paragraph is only partially relevant here and could probably be cut

If the reviewer agrees, we would like to keep this paragraph, since it summarizes and highlights the changes in the GLOF risk due to the observed glacier recession and expresses the urge for further monitoring.

395: 'The most extreme surface lowering. . .'

Corrected

Figure 3: caption should read 'example' not 'exemplary',

Corrected

but moreover I'm not sure what the value of the figure is since we can see most of this in Figure 1?

This figure indicates the glacier area recession at some example mountain ranges. In Figure 1 only one set of outlines is shown and no multi temporal area changes. In order to demonstrate the area changes on spatial scales (due to the scales not possible in Figure 1), this figure shows certain subsets (zoomed in) of the study region. We moved this figure to the supplemental material.

Figure 7: this caption needs some work I think. It took an age to work out that the red bars were vs the blue bars. How about 'Hypsometric distribution of measured glacier area with elevation (red) and total glacier area with elevation (blue), with mean dh/dt values in each elevation interval (blue dots). . .'?

Following the reviewer's suggestion, we revised the caption (also of the similar graphs in the supplemental material). We hope it is now more clear:

"Hypsometric distribution of measured (red bars) and total (light blue bars) glacier area with elevation in subregion R1 in the interval 2000-2016. Blue dots represent the mean $\Delta h/\Delta t$ value in each elevation interval. Error bars indicate NMAD of $\Delta h/\Delta t$ for each hypsometric bin. Grey areas mark the lower and upper 1% quantile of the total glacier area distribution. Black dashed line: mean glacier elevation; Red dashed line: equilibrium line altitude (ELA), see also Table S3. Area measurements are based on the glacier outlines from 2000, considering only regions with slopes below applied slope threshold (50°, see Section 4.2). Plots for other subregions are provided in the Supplementary material."

Response to the interactive comment on "Changes of the tropical glaciers throughout Peru between 2000 and 2016 – Mass balance and area fluctuations" by Thorsten Seehaus et al.

Christian Huggel (Referee)

**First of all we want to thank the reviewer for constructive comments on our manuscript. All comments have been taken into account and a list of answers and undertaken actions is given below. Answers are in blue font color**.

In my opinion this is a solid study with important new results and there is no doubt that I would like to see this paper published, eventually. While I think that the methods are good state of the art there are a few issues in the results which irritate me and make me wonder whether there are some more basic problems with data collection and analysis which I detail further below. To start I'm impressed by the amount of work done by the authors on a generally high level of data analysis, well presented. It adds new insights on glacier changes (area, surface changes and mass balance) which were previously not known on this level of detail and spatial coverage. I also like the discussion section which is transparent, comprehensive and encompasses the (full) coverage of available literature. In this discussion section the authors critically analyze a number of differences (and similarities) between their results and those of other studies. I can follow this discussion and I think it is mostly appropriate but I'm wondering whether there are underlying errors or uncertainties in terms of data sampling or analysis that may have gone undiscovered. I list here a number of possible problem areas:

The authors did not measure the full extent of the glaciers, and transparently report on it but the effect and possible uncertainties involved are not clear to me. The glacier area changes reported in Figure 9 and Table 2 contain numbers that raise some questions.
An area loss of only about 5% from 1970 to 2000 is in contradiction to what is generally reported, indicating values of 15-20% (Salzmann et al. 2013, Silverio and Jaquet 2004, others, incl. unpublished data). The authors indicate some aspects about incomplete inventories, or sampling issues. I'm not sure whether this large discrepancy can be explained by the mentioned aspects but urgently needs to be clarified.

The reviewer is right. An area change of -7% (Section 6.1) between 1970 and 2000 is too low. Therefore, it was mentioned in the text, that not all Cordilleras were mapped completely by the 1$^{st}$ Peruvian Glacier Inventory. In order to avoid irritations, we computed the changes considering only Cordilleras with full coverage in 1970 and revealed a value of -23% area changes (which fits to the findings by other studies). Consequently, we also adjusted Figure 9 and rephrased the statements in Section 6.1 and the caption of Figure 9.

"The comparison of our area measurement of 1916.6±48.3 km² in 2000 and the 1st Peruvian Glacier Inventory (Hidrandina SA, 1989) in 1970 (2041.85 km²) results in a retreat of -7% (-139.9 km²; 0.2% a$^{-1}$). However, the area changes amounts to -23%, considering only glaciated Cordilleras, which were completely mapped in the 1st Peruvian Glacier Inventory (UGRH, 2014). "

I'm also irritated by the error indications related to glacier area changes reported in Table 2, of up to 30% which is much higher than what is commonly achieved in remote sensing based mapping studies (ca. up to 5%). This also needs to be clarified.

The error values of the area changes dS in Table 2 result from basic error propagation using the uncertainties of the total glacier areas (Table 1 ~2.5-3.3%). Thus the uncertainty of the individual outlines is within the range of other studies. Area change computations are more sensitive to the

uncertainty of individual outlines, explaining the higher relative errors of up to 30%. The following statement was added in section 5.1, in order to clearly explain the error computation.

"It should be noted, that the uncertainty of the area changes (Table 2) result from the uncertainty of the individual inventories (quadratic sum), assuming independence of the individual area measurements."

I have seen many glacier mapping studies in the tropical Andes (published, or reviewed) which had errors because of inappropriately selected images with snow coverage which then resulted in erroneous glacier change results. I can't say whether this study is affected by a similar problem. In any case the authors should carefully review the literature they cite and whether some of these studies have such errors (at Coropuna for instance some published studies have such errors).

The reviewer is right, that temporary snow cover, but also clouds, can strongly impact the quality of glacier outlines. By selecting only images towards the end of the dry season, manual editing and inspection as well as cross-checking with high-resolution imagery (Google Earth), we tried to minimize this impact as far as possible. Particularly, the outlines in subregion R3 in 2016 might be affected by snow cover, since significant snow coverage was recently reported at the non-glaciated Cordillera Barroso during dry season 2016 (Léon et al., 2019), explaining the nearly stable glacier area between 2013 and 2016 in this region. A statement (see below) regarding this issue was added in Section 6.1.

"… A significant temporary snow cover at the non-glaciated Cordillera Barroso (subregion R3, close to the boarder to Bolivia) was observed during the dry seasons in 2015 and 2016 (Lèon et al. 2019), which fits to the suggested increased precipitation at high elevations during after 2013. Moreover, snowfall events during try season strongly affect the glacier albedo an thus lead to reduced ablation. However, the temporary snow cover during dry season 2016 in subregion R3 might have also affected the mapping of the glacier outlines. Albeit, only imagery with no or only minimal snow coverage is selected, it is quite likely that some remaining snow cover was located at the glaciated peaks, leading to a slightly larger glacier area in 2016 as compared to 2013. This bias is not quantifiable and certainly within the range of applied uncertainty. Thus, we conclude that the glacier area kept nearly stable after 2013 in subregion R3 and attribute this to the allocation of the remaining ice at higher altitudes and increased precipitation, especially during the dry season, even though a strong El Niño event occurred in this period. "

Regarding the quality of the cited studies: We checked the studies, especially at Coropuna. Some studies provide information potential impact due to snow cover (e.g. Peduzzi et al., 2010; Silverio and Jaquet, 2012), whereas other do not even provide information on the date of the accquisitions. Thus, it is difficult to assess the quality of the data.
Since, in this study comparisons with the general (average) trends of typically an ensembles of studies are discussed, we conclude that the impact of biases due to snow cover in other studies can be neglected.

I'm surprised by the drastic change of glacier and mass balance change of 2000-2013 vs 2013-2016. The authors list a number of plausible reasons, and I think the increased precipitation(accumulation) at high altitudes is a very important finding here. Nevertheless, important open questions remain. Fig. 9 indicates a change in the El Niño Index around the year 2013, changing from slightly negative to strongly positive. Reported mass balance and (high altitude) surface change can certainly be explained with this mechanism to some extent. But it is unclear (and not plausible) how precipitation changes would immediately translate into rather drastic changes in glacier area, even if the response of tropical glaciers through feedback processes including precipitation and albedo

changes is more direct than in mid-latitude glaciers. The comparison of their results with ground based mass balance measurements (glaciological method) are very significant.

Regarding the area changes: Other studies also inferred drastic changes in glacier area in correlation with El Niño. Silvero and Jaquet (2017) as well as Morizawa et al. (2013) reported clearly increased retreat during El Niño epochs and even area gain during La Niña. For example Silvero and Jaquet discovered a very high retreat rate of -23 km/a in Cordillera Blanca between 2014 and 2016 (average ONI 1.1, ~5%/a area loss, we discovered also 5%/a loss at subregion R1 for the period 2013-2016) and area gain of 5.24 km² (average ONI -0.20) between 1997 and 2002. Moreover, they inferred a linear relation between area changes and ONI with an R²=0.8. Additionally the glacier in the tropical Andes are in average the thinnest worldwide (Farinotti et al. 2019). Thus, increased ablation will cause more pronounced area reduction as compared to other regions. We conclude that the El Niño conditions and the associated increased ablation in the period 2013-2016 can be attributed as the driver for the observed increased recession. A comparison of our area change measurements with field measurements is not meaningful due to the large differences in the basin delinations.
The following statemant was added Section 6.1:
"This pattern fits also to the finding by Morizawa et al. (2013) (at Condoriri Glacier, Bolivia) and Silvero and Jaquet (2017) (at Cordillera Blanca, subregion R1). Both studies reported enhanced recession during El Niño events and even area gains during La Niña epochs. The latter study also discovered a linear relation between glacier retreat and ONI (R²=0.8) and reports an change rate of -5% a$^{-1}$ for 2014-2016 that is equal to our change rate at subregion R1 for 2013-2016. Moreover, the glaciers in the Tropical Andes are in average the thinnest worldwide (Farinotti et al. 2019). Therefore, the increased melt will lead to more pronounced changes in glacier area as compared to other glacier region. "
Regarding glaciological mass balance measurements, see answer to next comment.

The authors are right that there are problems with the mass balance measurements which in fact are very challenging on these glaciers. Nevertheless, the authors should investigate this issue in more depth. I would also recommend to look in more detail on locally available field data which co-author Alejo Cochachin disposes of. The measurement interval (2000-2013) could have an effect, and changes towards more negative glacier mass balances could have been started earlier than 2013.

We carried out an extended analysis of the field measurements. The glaciological mass balances and ELA values are correltated with annual ONI values (Sep.-August). A clear trend towards higher mass losses and ELA values is visible for positive ONI values, indicating El Nino conditions (Fig. S30). Moreover, the annual mass balances at Yanamarey and Artesonraju, show strongly negative mass balances for 2015/16 and increase average mass loss trends after 2013. Unfortunatly, the mass balance series do not cover our whole remote sensing observation period, starting in 2000. Since, the mass balance variability of tropical glacier is strongly dominated by the lowest section (Soruco et al., 2009), we analyses the geodetic mass balances at regions below 4900/5000 m a.s.l. (approx. average ELA) separatly. Moreover, these lower sections have the highest density of glaciological point measurements. For both glacier, an onset of the rapid surface lowering at the terminus region is obvious already before 2013 (Fig. S31, 2011/12 for Artesonraju, unfortunatly the data at Yanamarey had some issues in the period 2009-2011). A comparison with the remote sensing data for the period 2013-2016 shows at Yanamarey a good agreement (average glaciological: -3152 kg/(m²a), geodetic: -3164 kg/(m²a)), whereas at Artesonraju the values differ (average glaciological: -4206 kg/(m²a), geodetic: -2696 kg/(m²a)). This difference can be partly attributed to the different glacier basin definitions. To sum it up, we revealed higher mass loss rates after 2013 and for positive ONI values a trend towards more negative mass balances (glaciological data). These, findings fit to our observed geodetic mass balance trends. Moreover, we discovered a strong increase in the mass loss for the lowest glacier section, starting around 2011.

The following text was added to the manuscript in Section 6.2:
"The correlation of the annual glaciological measurements with the average ONI of the respective observation periods indicates a trend towards increased mass loss and higher ELA during El Niño conditions (Figure S30). These tendencies fit to the observations by Silvero and Jaquet (2017) and Morizawa et al. (2013) (see Section 6.1) and support our revealed correlation between the increased glacier wastage in the period 2013-2016 and the strong El Niño event during this period.
Since the highest density of glaciological point measurements are collected at the lowest section at both glacier and considering the observation of Soruco et al. (2009) at Zongo Glacier, Bolivia, that the terminus region strongly dominates the mass balance variations of a tropical glacier, we did an analysis of the glaciological mass balance at regions below 4900 m a.s.l. and 5000 m a.s.l. at Yanamarey and Artesonraju glaciers, respectively. A trend toward increased mass losses after 2011 is obvious (Figure S31, unfortunately the data at Yanamarey Glacier for the period 2009-2011 was incomplete), fitting to the revealed more negative geodetic mass balances after 2013. Moreover, the most negative values are derived for the period 2015-2016, which also supports our suggestion, that ENSO force the increased glacier wastage after 2013. The comparison between the geodetic and the average glaciological measurements at the terminus regions in the period 2013-2016 revealed a good agreement of both methods at Yanamarey Glacier (average glaciological: -3152 kg m$^{-2}$a-1, geodetic: -3164 kg m$^{-2}$a-1), whereas at Artesonraju Glacier, the geodetic measurements indicate higher mass loss (average glaciological: -4206 kg m$^{-2}$a-1, geodetic: -2696 kg m$^{-2}$a-1). This difference can be partly attributed to the different glacier basin definitions and slightly different observation intervals, but also to limitations of the individual methods, as discussed above."

Also, just as an additional information, according to mass balance measurements we did in collaboration with the Peruvian colleagues indicate that mass balances (since 2010) are much more negative in the Cordillera Blanca than in the Cordillera Vilcanota.

We appreciate this information. I sent requests regarding mass balance measurements in the Cordillera Vilcanota region to the respective institution. However, I did not receive an answer until now. If this data will become available during the further review process, I will certainly include it in the analysis.

All these points, open questions and uncertainties leave me with considerable doubts whether there are (basic?) problems with data collection, processing and analysis. For me these are the fundamental points that absolutely need to be clarified before this study can be published. I encourage the authors to do a serious investigation about these issues such that we can have reasonable confidence that the reported results reflect the reality and are not distorted by any errors.

References:

[revised manuscript text omitted]

---

## Editor Decision (ED1)

Dear authors,

Thank you very much for your responses and the changes to the manuscript. I am happy to accept your revised manuscript subject to technical corrections, which are reported before.

1) I strongly encourage the authors to carry out a thorough proof-reading and correction of the revised text, as the new sections contain English and grammar errors and sentences that read awkward or not polished enough. I provide here few examples:

_New text in section 6.1
_"**an** change rate"
_"significant temporary snow cover at the non-glaciated Cordillera Barroso (subregion R3, close to the boarder to Bolivia) was observed during the dry seasons in 2015 and 2016 (Lèon et al. 2019), which fits to the suggested increased precipitation at high elevations **during after** 2013" - please correct the "during after"

_Line 380: "on average" instead of "in average"

_Please check your use of commas, such as on line 294: "It should be noted, that the uncertainty… ". I think a comma here, and in many other instances, is not necessary.

These are just examples and I request the authors to check all the text for spelling and English/grammar errors, as well as style.

_The new text added in response to the comment by Reviewer C. Huggel on the very small area change reported is not clear – the text in your response is clearer instead.

"The comparison of our area measurement of 1916.6±48.3 km² in 2000 and the 1st Peruvian Glacier Inventory (Hidrandina SA, 1989) in 1970 (2041.85 km²) results in a retreat of -7% (-139.9 km²;
0.2% a-1). However, the area changes amounts to -23%, considering only glaciated Cordilleras, which were completely mapped in the 1st Peruvian Glacier Inventory (UGRH, 2014). " ""

I suggest the authors change this into:

"However, not all Cordilleras were mapped completely by the 1st Peruvian Glacier Inventory. When considering only the glaciERISED Cordilleras which were completely mapped in the 1st Peruvian Glacier Inventory (UGRH, 2014), the area changes amount(NO s) to -23%".

The text should be improved by proof-reading by a native speaker for style and errors, and I strongly suggest the authors do this before resubmitting their final manuscript.

e.g.: "Thus, we conclude that the glacier area kept nearly stable after 2013 in subregion R3 and attribute this to the allocation…."

Please use a better expression for "the glacier area kept stable".

2) Since you have added this sentence in the abstract:

The mapping of debris-covered glacier extents is supported by SAR-coherence information

Following a suggestion by reviewer D. Quincey, please add the corresponding information also in the Methods section.

---

## Author Response (AR2)

Dear authors,
Thank you very much for your responses and the changes to the manuscript. I am happy to accept your revised manuscript subject to technical corrections, which are reported before.

Dear Editor,
thank you very much for accepting the manuscript. We applied the suggested corrections (see answers to comments below) and sent it to a proof-reading agency.
We hope the manuscript is now ready for further processing.

1) I strongly encourage the authors to carry out a thorough proof-reading and correction of the revised text, as the new sections contain English and grammar errors and sentences that read awkward or not polished enough. I provide here few examples:
_New text in section 6.1
_"an change rate"
_"significant temporary snow cover at the non-glaciated Cordillera Barroso (subregion R3, close to the boarder to Bolivia) was observed during the dry seasons in 2015 and 2016 (Lèon et al. 2019), which fits to the suggested increased precipitation at high elevations during after 2013" - please correct the "during after"
_Line 380: "on average" instead of "in average"
_Please check your use of commas, such as on line 294: "It should be noted, that the uncertainty... "
I think a comma here, and in many other instances, is not necessary. These are just examples and I request the authors to check all the text for spelling and English/grammar errors, as well as style.

We applied the highlighted corrections and proof-reading of the whole manuscript by an native speaker was done as well.

_The new text added in response to the comment by Reviewer C. Huggel on the very small area change reported is not clear – the text in your response is clearer instead.
"The comparison of our area measurement of 1916.6±48.3 km2 in 2000 and the 1$^{st}$ Peruvian Glacier Inventory (Hidrandina SA, 1989) in 1970 (2041.85 km2) results in a retreat of -7% (-139.9 km2; 0.2% a-1). However, the area changes amounts to -23%, considering only glaciated Cordilleras, which were completely mapped in the 1st Peruvian Glacier Inventory (UGRH, 2014). "
I suggest the authors change this into:
"However, not all Cordilleras were mapped completely by the 1 st Peruvian Glacier Inventory. When considering only the glaciERISED Cordilleras which were completely mapped in the 1st Peruvian Glacier Inventory (UGRH, 2014), the area changes amount(NO s) to -23%".

We changed the text according to your suggestion.

The text should be improved by proof-reading by a native speaker for style and errors, and I strongly suggest the authors do this before resubmitting their final manuscript.
e.g.: "Thus, we conclude that the glacier area kept nearly stable after 2013 in subregion R3 and attribute this to the allocation...."
Please use a better expression for "the glacier area kept stable".
See comment above.

2) Since you have added this sentence in the abstract:The mapping of debris-covered glacier extents is supported by SAR-coherence information
Following a suggestion by reviewer D. Quincey, please add the corresponding information also in the Methods section.

Information regarding the SAR-Coherence analysis can be found in section 4.1 (line 165-175)

Moreover, I am currently preparing the submission of the data to PANGAEA, WGMS and GLIMS. Maybe, the respective DOI can be included during the next processing step of the paper.

Kind regards
Thorsten Seehaus, on behalf of all authors

[revised manuscript text omitted]

**10    Tables (place holder, see other document)**

585

Table 1. area

Figure 2: slope-dh

Table 2. dS dM

Figure 4: polar ds

Figure 5: dh 00-13

Figure 6: dh 13-16

Figure 7: hypso-dh

Figure 8: temp. evol.

Figure 9: polar mb